# The Role of Potential Oxidative Biomarkers in the Prognosis of Acute Ischemic Stroke and the Exploration of Antioxidants as Possible Preventive and Treatment Options

**DOI:** 10.3390/ijms24076389

**Published:** 2023-03-28

**Authors:** Fatima Zahra Kamal, Radu Lefter, Hassna Jaber, Ioana-Miruna Balmus, Alin Ciobica, Alin-Constantin Iordache

**Affiliations:** 1Laboratory of Physical Chemistry of Processes, Faculty of Sciences and Techniques, Hassan First University, B.P. 539, Settat 26000, Morocco; 2Biomedical Research Center, Romanian Academy, Iași Branch, 8th Carol I Avenue, 700506 Iași, Romania; 3Laboratory of Natural Resources and Sustainable Development, Department of Biology, Faculty of Sciences, Ibn Tofail University, Kenitra 14000, Morocco; 4Department of Exact Sciences and Natural Sciences, Institute of Interdisciplinary Research, “Alexandru Ioan Cuza” University of Iasi, 26th Alexandru Lapusneanu Street, 700057 Iasi, Romania; 5Department of Biology, Faculty of Biology, Alexandru Ioan Cuza University of Iasi, 20th Carol I Avenue, 700506 Iași, Romania; 6Faculty of Medicine, “Grigore T. Popa”, University of Medicine and Pharmacy, Strada Universitatii 16, 700115 Iasi, Romania

**Keywords:** ischemic stroke, oxidative stress, oxidative stress biomarkers, antioxidant therapy

## Abstract

Ischemic strokes occur when the blood supply to a part of the brain is interrupted or reduced due to arterial blockage, and it often leads to damage to brain cells or death. According to a myriad of experimental studies, oxidative stress is an important pathophysiological mechanism of ischemic stroke. In this narrative review, we aimed to identify how the alterations of oxidative stress biomarkers could suggest a severity-reflecting diagnosis of ischemic stroke and how these interactions may provide new molecular targets for neuroprotective therapies. We performed an eligibility criteria-based search on three main scientific databases. We found that patients with acute ischemic stroke are characterized by increased oxidative stress markers levels, such as the total antioxidant capacity, F2-isoprostanes, hydroxynonenal, total and perchloric acid oxygen radical absorbance capacity (ORAC_TOT_ and ORAC_PCA_), malondialdehyde (MDA), myeloperoxidase, and urinary 8-oxo-7,8-dihydro-2′-deoxyguanosine. Thus, acute ischemic stroke is causing significant oxidative stress and associated molecular and cellular damage. The assessment of these molecular markers could be useful in diagnosing ischemic stroke, finding its causes, predicting its severity and outcomes, reducing its impact on the cellular structures of the brain, and guiding preventive treatment towards antioxidant-based therapy as novel therapeutic alternatives.

## 1. Introduction

If considered independently from other cardiovascular diseases, strokes are the fifth most prevalent cause of mortality worldwide [1], with a global increase of around 1.9 million cases between 2013 and 2019 [2,3]. According to their pathogenic mechanisms, strokes are commonly divided into two subtypes: ischemic and hemorrhagic [4]. Approximately 80% of all strokes include ischemic events, with an incidence rate of 7.63 million in 2019 [5]. The life-threatening character of strokes renders them major medical emergencies, and treatment must be administered without delay. This may be one of the factor that makes it difficult to collect data from human subjects and observe the pathogenic mechanisms that occur immediately before and after the stroke. The use of animal models has made it possible to mimic the stroke processes, allowing for the description of pathological mechanisms, risk factors, and potential management options. As a result, researchers have valuable information on ischemic stroke.

Based on the multicenter Trial of Org 10172 in Acute Stroke Treatment (TOAST) Classification System, the ischemic stroke subtypes are categorized as follows: (1) cardioembolism (linked to cardiac dysrhythmias, valvular heart disease, and left ventricle thrombi), (2) large-artery atherosclerotic, (3) lacunar (microatheromatosis), (4) other specific etiology (dissections, vasculitis, specific genetic disorders, and others), and (5) strokes of unknown etiology [6,7].

The risk factors for the ischemic stroke are usually divided into non-modifiable risks (age, sex, race, or ethnicity, genetic predisposition, and low weight at birth), and modifiable risk factors (diabetes mellitus, alcohol abuse, obesity, hypertension, smoking, metabolic syndrome, dyslipidemia syndrome, and others) [5]. According to the global data published by *Lancet Neurology*, 90.5% (95% UI 88.5–92.2%) of the stroke burden is attributable to modifiable risk factors, with 74.2% (95% UI 70.7–76.7%) being related to behavioral risk factors (such as smoking, poor diet, and low physical activity) [8,9]. Thus, the risk for developing an ischemic stroke is higher for people that accumulate chronic conditions that predispose them to high blood pressure, defects in the coagulation process, or blood vessels stenosis (of atherogenic or inflammatory origin): cardiovascular diseases, metabolic diseases, viral infections, psychiatric conditions (affective disorders, or disorders due to exposure to stress), and other pathological conditions that indirectly influence the occurrence of ischemic stroke events (sleep apnea, migraines, and hereditary blood coagulation defects). Due to the extent of the damage often produced by the ischemic strokes, the prognosis and outcomes are frequently complicated by serious physical, psycho-intellectual, functional, and socio-professional issues, such as fatigue, motor disability, depression, activity limitations, anxiety disorders, and an increased risk of re-hospitalization and institutionalization [10], all of which put a great burden on the patients and their families [11]. In this context, prevention, reduction of exposure to risk factors, fast and efficient diagnosis, and good management of ischemic strokes’ consequences are critical.

Based on the current knowledge about the pathological mechanisms of ischemic strokes, diagnosis is carried out by imaging (head computer tomography followed by angiography) [12], while treatment efficiency and recovery are screened using an interdisciplinary approach, as many of the life and health aspects of the patients are altered during an acute ischemic stroke. However, early or mild ischemic strokes could not be seen on imaging tools [13]. During the diagnosis and first counteracting measures, speed is crucial, and thus any molecular test that could ensure fast and non-invasive detection of ischemic stroke could be of extreme aid. In this way, many biochemical biomarkers have been previously described, but they are not entirely useful due to lack of specificity or the timespan gaps in some protein biomarkers dynamics (especially those related to changes that occur after thrombolytic therapy) [13,14]. However, acute ischemic stroke pathophysiology has multiple components, and molecular biomarkers, such as thromboembolic component, inflammatory component, oxidative stress, and metabolic component, could be found in any of them [15,16,17].

Due to the production of reactive oxygen species (ROS), oxidative stress has been suggested as one of the main mechanisms of physiological damage and disease progression in several diseases. Stroke is an ideal example of the disrupted oxidant-antioxidant homeostasis consequences in humans [18]. Alongside the obstruction of blood flow (ischemia), reperfusion induces severe neuronal damage in acute ischemic stroke. Sever-al research studies have recently established evidence in favor of the implication of oxidative stress in reperfusion injury following recanalization [19]. The brain cells are naturally predisposed to oxidative damage due to their high polyunsaturated fatty acid content. Other factors, such as low antioxidants levels and high pro-oxidant levels (iron), together with high oxidative metabolism rates, were found to further increase the severity of oxidative damage during stroke-related injury [20]. Studies on the acute ischemic stroke pathogenesis have highlighted that disrupted calcium homeostasis leading to increased calcium release in the brain activates ROS generation pathways that create uncontrollable bulges of oxidant levels within the brain tissues [21]. ROS are also responsible for the inactivation of many functional enzymes (due to protein oxidation) and alteration of carbohydrates that lead to apoptotic signaling assistance [19]. Consequently, Saver [22] estimated that 1.9 million neurons, 14 billion synapses, and 7.5 miles of myelinated fibers are destroyed each minute during an acute ischemic stroke.

A link between energy metabolism depletion, oxidative stress, and ischemic stroke has been proposed. The interruption of blood flow to the brain contributes to defective energy metabolism in the pathophysiological hallmark of ischemic stroke. This impaired energy metabolism initiates the cascade of oxidative stress injury characterized by an oxidant/antioxidant imbalance in the body, leading to excessive production of ROS and hydroxyl radicals that result in the brain damage which follows a stroke [23,24].

During the ischemic stroke initiation, glucose, and oxygen deprivation lead to cellular ROS production increase via mitochondrial depolarization and complex IV inhibition, which further leads to upstream accumulation of reduced compounds, such as O^2−^ [25]. Furthermore, the acidic environment caused by hypoxia further accelerates the conversion of O^2−^ into H_2_O_2_ or into the more reactive hydroxyl radical (HO^−^) [25]. In addition, the elevated levels of extracellular glutamate cause excitotoxicity by over-activating various glutamate receptors of neurons, such as the N-methyl-D-aspartate (NMDA) receptors, contributes to an overload in calcium influx; the latter causes neurons oxidative stress, aggravates mitochondrial dysfunction, and activates cellular proteases and lipases [26]. NMDA receptors also trigger nitric oxide synthase (NOS), which is the enzyme that catalyzes the production of NO from L-arginine, causing an increase in ONOO^−^ production [27].

Besides mitochondria, xanthine oxidase (XO) and NADPH oxidase are two elementary sources of superoxide anion, a key radical after ischemic stroke. XO is a molybdo-flavin enzyme that catalyzes the oxidation of hypoxanthine to xanthine and xanthine to urate, which is the process that reduces molecular oxygen and produces O_2_^−^ [28]. There are two isoforms of this enzyme: xanthine dehydrogenase (NAD dependent dehydrogenase) and XO (oxygen dependent superoxide production oxidase) [29]. Ischemia induces acute ATP depletion, which leads to the accumulation of hypoxanthine and xanthine, substrates for the ROS-producing XO [30]. In gerbils, these products contributed to brain edema caused by ischemia/reperfusion [31]. Several studies also reported that, in a rat model of acute ischemic stroke, XO expression on the infarcted side was significantly higher than that on the non-infarcted side and continued to increase over time [32,33]. NADPH is a significant additional source of ROS production in ischemic stroke [33]; NADPH oxidase (NOX) is an enzyme complex on the cell membrane responsible for ROS production following ischemic stroke [34]. NOX is made up of six subunits (NOX1-NOX5, and Dual Oxidase 1 and 2) that generate superoxide via one-electron trans-membrane transfer to molecular oxygen [35]. After an ischemic stroke, an increase in NOX4 and NOX2 expression was observed [36,37]. NOX2 was found to stimulate the neuronal ROS generation and to be the primary source of N-methyl-D-aspartate receptor activated superoxide production [38].

The other three major enzymatic pathways that catalyze the release of ROS are cyclooxygenases, lipoxygenases, and cytochrome P450 enzymes. During ischemia, these enzymes are involved in the conversion of free arachidonic acid released from cell membrane phospholipids by phospholipases, especially phospholipase A2 into eicosanoids. These reactions generate reactive oxygen species (ROS), such as superoxide anion (O_2_^−^) and HO^−^ [39] (Figure 1).

It was recently documented that even a single stroke event could lead to permanent disability, while its recurrence could be fatal due to multiple thrombus formation and arterial re-occlusion [6,40,41]. Hence, understanding the underlying mechanisms of stroke is crucial in reducing stroke-associated mortality. Because the association of ROS with the pathogenesis of stroke is well established, it is currently thought that redox biomarkers are an important tool for monitoring the progression of stroke and its prognosis [42,43]. However, it is important to understand that the variance of redox biomarkers levels is seen in many diseases, which makes it difficult to associate one specific biomarker with acute ischemic stroke. Concurrently, the changes occurring in oxidative homeostasis can be carried out by qualitatively and quantitatively evaluating the biomarkers that enable the accessible disease prognosis [21,44,45,46,47]. The oxidative stress-associated tissue damage occurring in many pathologies is often appraised by determining the extent of the cellular pathways that compel the structural and functional impairment of molecules. These modifications occur in stages where the end products are the most stable and cause the most harm to the tissues due to complete loss of function.

To date, specific correlations between various redox biomarkers and the incidence of stroke events have been successfully established, and lipids, DNA, enzymes, and protein oxidation are considered to be possible biomarkers of ischemic stroke. However, there is little evidence regarding the implication of these oxidation products in ischemic stroke development [19,21,48]. As immediate intervention during a stroke is ethically necessary, relevant biomarkers of oxidative stress have been reported solely in blood samples collected following the thrombolytic therapy. Therefore, there is limited understanding regarding the role of these specific biomarkers in the events occurring just before or during a stroke event [21,48,49]. Furthermore, it was previously suggested that oxidative stress extent could compel prognosis and outcomes following an acute ischemic stroke related to cognitive impairments and mortality risk [49]. For an instance, Wang et al. [50] found that the correlation between oxidized LDL plasma levels and cognitive performance following stroke (as evaluated by Mini-Mental State Examination test) fit into a linear regression model. The same group also reported that stroke patients with increased oxidized LDL plasma levels had higher risk of death or poor functional outcomes at 1 year after a stroke event when the event was related to large-artery atherosclerosis [51,52]. In this context, the use of oxidative stress biomarkers in acute ischemic stroke diagnosis and prognosis could potentially offer additional information about the injury extent or short or long-term consequences.

Thus, in this review, we aimed to evaluate the potential use of the main oxidative stress biomarkers in acute ischemic stroke diagnosis and prognosis, as well as their efficiency to predict acute ischemic stroke short and long-term outcomes (response to treatment and recovery. As oxidative stress was previously demonstrated as a major component in acute ischemic stroke pathophysiology, we also aimed to describe the possible use of antioxidant therapy in management of this condition.

## 2. Lipid Peroxidation

The oxidative damage to lipids is assessed by quantitatively or qualitatively estimating the level changes of malondialdehyde (MDA), 4-hydroxy-2-nonenal (HNE), oxidized LDL, and F2-isoprostanes in the relevant body fluids (blood, cerebrospinal fluid, or others). These biomarkers are also used in studying the triggers in stroke-related thrombotic events. In general, lipid peroxidation is mainly addressing the lipid molecules within the cellular membrane structures and leads to lipid free radicals and dienic hydroperoxides production. Both these compounds are highly toxic, and the latter is an unstable compound that is decomposed into dienals, aldehydes, or alkanes [53]. One lipid peroxidation process was described in cellular membranes as a result of iron implication in oxidative damage in cellular membranes (ferroptosis) [54]. Ferroptosis is an iron dependent form of non-apoptotic programmed cell death that is caused by the harmful accumulation of lipid-based reactive oxygen species due to the failure of the complex lipid peroxide repair systems, such as glutathione-GPX4 [55]. Despite the as yet unknown details of this very different cellular death process, ferroptosis was already described in ischemic stroke as a potential therapeutic target [56,57]. The possible mechanism of ferroptosis in ischemic stroke was associated with glutathione peroxidase 4 inhibition, Fenton reaction of Fe^2+^ in the presence of hydrogen peroxide, and lipid peroxidation [56,57].

### 2.1. Malondialdehyde

MDA is a major end product of lipid peroxidation (during the metabolism of polyunsaturated fatty acids and arachidonic acid) formed due to cellular damage and consequent degradation of polyunsaturated fatty acids from the membrane phospholipids molecules [58]. MDA could be produced in the cells via multiple pathways. Other than the lipid peroxidation pathway, the main pathway in which MDA is formed in the animal tissues is thromboxane and prostaglandin biosynthesis [59].

Among the different thiobarbituric acid reactive substances (TBARs), MDA is the most abundant, making it the most important lipid peroxidation biomarker assessed from blood samples by the TBARs method [60]. TBARs are common peroxidation products of lipids, and Tsai et al. reported that serum TBAR levels on day 7 can be sensitive predictors of related complications that occur after ischemic strokes [61]. However, their use is not preferred in any disease because they lack specificity.

MDA is highly reactive, and thus, it further attacks proteins and nucleic acids and causes substantial damage to cellular constituents [59]. As it could actively interact with lipoproteins, MDA is one of the most relevant biomarkers in atherosclerotic diseases and in evaluating the predisposition to cardiovascular risk [60]. The mutagenic potential of MDA was previously reported by its participation in mutagenic DNA adducts formation (2-aminopyrimidines are specifically formed when MDA reacts with deoxyadenosine and deoxyguanosine) [62]. Because of these functions, MDA is currently the most evaluated lipid peroxidation biomarker.

Regarding the MDA association with ischemic stroke, there are several recent studies which suggest that MDA bloodstream levels could be positively correlated with stroke incidence and the outcome of neurological function alteration [42,63]. The pathways in which MDA is produced in acute ischemic stroke are isolated to the prostaglandin and oxidative stress-related processes. As thromboxane A2 is implicated in post-ischemic-stroke thrombotic events, MDA excess production could also originate from this pathway [64].

Thus, the detection of MDA is not restricted to the assessment of blood samples, but can also use saliva samples; saliva may even offer better accuracy (92% in saliva versus 81% in blood samples). Al-Rawi et al. [65] suggested that the assessment of salivary MDA could be a relevant marker in identifying patients with high risk for stroke. A rapid non-invasive test could predict the risk for an important life-threatening event, even though this end product may not be disease specific (Table 1).

In another study, compared to healthy individuals, MDA levels were reported to be significantly higher in 100 ischemic stroke patients with simultaneous low levels of total antioxidant power (TAP). However, the study did not report any association between oxidative stress and stroke severity [45]. It could therefore be important to identify the pathway through which MDA is involved in ischemic stroke development, as it can also be seen in Figure 2.

### 2.2. F2-Isoprostanes and 8-Iso-Prostaglandin F2 Alpha

F2-isoprostanes are another example of lipid peroxidation products. These products are prostaglandin-like stable compounds formed during the peroxidation of arachidonic acid and membrane phospholipids [66,67]. The main process that contributes to the formation of F2-isoprostanes is the auto-oxidation of membrane lipids. However, when oxidative stress occurs (in acute ischemic stroke and other conditions involving oxidative stress and inflammation), ROS (H_2_O_2_) could participate in forming F2-isoprostanes and 8-iso-prostaglandin F2 alpha through arachidonic acid oxidation [68,69].

The studies that reported the use of F2-isoprostanes in the early prognosis of ischemic stroke are rather scarce compared to those examining MDA (Table 1). Lorenzano et al. [46] suggested that F2-isoprostanes levels can independently predict the infarct growth and volume in acute ischemic patients. In another study, the same group showed the use of F2-isoprostane in predicting the risk for tissue infarction in patients with acute ischemic stroke [70]. An earlier study also reported F2-isoprostanes to be one of the most reliable biomarkers in post-stroke conditions as well as a predictor of antioxidant therapy efficiency [71]. However, although F2-isoprostanes are suitable biomarkers for predicting stroke outcome in general, they are unreliable because arachidonic acid can auto-oxidate in absence of stress.

Another option, 8-iso-prostaglandin F2 alpha (8-isoPGF2α; isomer of prostaglandin 2α), is considered more reliable due to higher specificity, sensitivity, and stability [69]. In acute ischemic stroke, 8-isoPGF2α is mainly produced when ischemia occurs, in platelet recruitment via GpIIb/IIIa activation. Thus, ROS-derived 8-iso-PGF2α is thought to signal thrombosis development that follows an acute ischemic stroke [72]. However, even though it is secreted in urine, making it much easier for sample collection and detection in laboratories, 8-isoPGF2α occurs after reperfusion in some cases, which is rather late [73].

### 2.3. 4-Hydroxy-2-nonenal

4-Hydroxy-2-nonenal (HNE) is primarily produced during the peroxidation of omega 6 polyunsaturated fatty acids. Similar to F2-isoprostanes, there is less evidence regarding the association of HNE levels with ischemic stroke risk [74,75]. However, since HNE concentration can be determined based on immunological techniques using anti-HNE antibodies, it is preferred over MDA detection (Table 2) [72]. A strong relation between HNE and brain tissue oxidative stress has been reported by Lee et al. [74] in a mice model of brain ischemic stroke. It was reported that middle cerebral artery occlusion after HNE administration significantly contributed to ischemia-induced infarction area infraction. Convincing evidence regarding the potential role of HNE in ischemic stroke was also provided by Guo et al. [75], who used aldehyde dehydrogenase 2 (ALDH2)-deficient stroke-prone spontaneously hypertensive rats as experimental models to prove that ALDH2 activation leads to HNE catabolism, which provides neuroprotection. Through comparison to the animal models findings, an 8-year prospective study showed that all the ischemic stroke patients developed increased HNE levels, compared to non-ischemic stroke subjects [75] (Table 1). Moreover, the increased HNE levels persisted in the patients’ bloods for over 6 months [75].

### 2.4. Oxidized LDL

Oxidized LDL is also an important biomarker of lipid peroxidation [76]. When there is excess ROS, LDL molecules could be mildly oxidized to trigger inflammatory cytokines synthesis. If the lipid peroxidation process continues, LDL molecules could become highly oxidized to trigger macrophages activation and apoptosis [77,78,79]. Their main implication in pathological mechanisms remains the atherogenic processes, with no exception for ischemic stroke in which oxidized LDL is produced by a vicious cycle consisting of ROS production by the oxidized LDL-activated platelets [80,81,82].

Due to their high reactivity, oxidized LDLs were demonstrated to disrupt endothelial function. They also promote platelet aggregation and leukocyte-endothelial cell adhesion [83,84]. Consequently, they are potent triggers of thrombotic events and could further increase thrombosis and recurrent ischemic stroke risks in stroke patients [84]. Wang et al. [50,51,52] studied the association between oxidized LDL and ischemic stroke outcomes in 3688 stroke patients. The study reported a high risk of death and poor functional outcome within 1 year after stroke onset in patients with high levels of oxidized LDLs. Meanwhile, Yang et al. [85] reported that the decrease in oxidized LDL levels is associated with better outcomes and recovery in Ischemic stroke patients (Table 1).

**Table 1 ijms-24-06389-t001:** Oxidative stress biomarkers levels in acute ischemic stroke.

Biomarkers	Source Type	Study Design	Concentration	Comment	References
** *Lipid peroxidation* **					
MDA (µmol/L)	Saliva	150 individuals:		Salivary and serum MDA levels were significantly higher in the ischemic stroke group than the healthy control and risk group.	[65]
		Ischemic stroke, N = 50	0.64 ± 0.22		
		Healthy control, N = 25	0.23 ± 0.07		
		Risk group, N = 75:			
		Hypertension	0.75 ± 0.21		
		Type 2 diabetes	0.65 ± 0.22		
		Ischemic heart disease	0.48 ± 0.13		
	Serum	150 individuals:			
		Ischemic stroke, N = 50	2.51 ± 1.11		
		Healthy control, N = 25	1.12 ± 0.35		
		Risk group, N = 75:			
		Hypertension	2.19 ± 0.83		
		Type 2 diabetes	2.39 ± 0.97		
		Ischemic heart disease	2.22 ± 0.73		
	Blood	200 individuals:		MDA was significantly higher in stroke patients than in healthy controls.	[45]
		Ischemic stroke, N = 100	7.11 ± 1.67		
		Healthy control, N = 100	1.64 ± 0.82		
4-HNE (µmol)	Plasma	60 men:		The plasma 4-HNE concentrations in patients with ischemic stroke were higher than those in healthy control.	[74]
		Ischemic stroke: N = 24,	≈11		
		Normal men, N = 36	≈9		
TBARs (µmol/L)	Serum	180 individuals:		The concentration of TBARS was significantly higher in stroke patients than in the controls.	[61]
		Acute ischemic stroke, N = 100:			
		Small-vessel, N = 75	20.7 ± 2.6		
		Large-vessel, N = 25	19.7 ± 1.2		
		Healthy control (age- and sex-matched), N = 80	≈16		
** *DNA oxidation* **					
8-oHdG (ng/mgCr)	Urine	44 acute ischemic stroke patients: Lacunar, N = 9 Atherothrombotic, N = 22 Cardioembolic, N = 13	Day 0 15.8 ± 6.9 12.8 ± 11.7 11.8 ± 5.6 Day 7 16.1 ± 5.1 16.2 ± 15.5 13.0 ± 5.0	8-oHdG urinary levels increase in time, following AIS. In patients with better outcomes, the level increase is significantly slower.	[86]
8-oHdG (ng/L)	Blood	241 acute ischemic stroke patients:		There was a difference between 8-oHdG levels and ischemic stroke severity in depressed versus non-depressed patients. Mild positive Spearman correlation between 8-oHdG levels and catalase activity. Urinary 8-oHdG levels could be used as reliable and valuable biomarkers to predict functional outcomes in stroke rehabilitation.	[87]
		Depressive post-ischemic stroke, N = 70	218.0 (170.6–246.7)		
		Non-depressive post-ischemic stroke, N = 171	164.8 (121.1–208.0)		
8-oHdG (ng/mg creatinine)	Urine	Acute ischemic stroke patients, N = 61:		There was a difference between 8-oHdG levels and ischemic stroke severity in depressed versus non-depressed patients. Mild positive Spearman correlation between 8-oHdG levels and catalase activity. Urinary 8-oHdG levels could be used as reliable and valuable biomarkers to predict functional outcomes in stroke rehabilitation.	[88]
		Before rehabilitation	5.87 ± 2.77		
		After rehabilitation	5.60 ± 2.47		
** *Protein oxidation* **					
Protein carbonyls (nmol/mg protein)	Plasma	163 individuals:		Protein carbonyls were not significantly different in the experimental groups compared to the controls.	[48]
		Healthy control (gender and age matched), N = 81	0.25 ± 0.04		
		Acute ischemic stroke, N = 82	0.28 ± 0.04		
Homocysteine (µmol/L)	Plasma	653 individuals:		The increase in total homocysteine concentrations was associated with a 6% to 7% increase in stroke risk	[89]
		Stroke patients (male and female), N = 120	≥18.6		
		Control subjects, N = 533	12.0		
	Serum	225 individuals		Moderate hyperhomocysteinema has been proposed as an independent risk factor for stroke in middle-aged British men	[90]
		Stroke patients (male), N = 107	13.7		
		Healthy subjects (male), N = 118	11.9		
	Blood	71 individuals:		Hyperhomocysteinemia is independent risk factors for stroke	[91]
		Ischemic stroke patients (47 males, 24 females), N = 71	22.76 × 104 ± 12.67		
Glutathion (nmol/g of brain tissue)	Brain tissue	36 White male rats (weighing 260–300 g; 2–3 months of age)		GSH homeostasis as an oxidative stress marker has been disturbed in global and focal ischemia. After focal and global cerebral ischemia, a significant drop was recorded in the levels of the reduced forms of GSH in blood plasma. This effect may be attributed to their oxidation.	[92]
		Control:			
		Reduced GSH	592 ± 28		
		Oxidized GSH	38 ± 4		
		Bilateral occlusion of the common carotid arteries (BCAO):			
		Reduced GSH	575 ± 24		
		Oxidized GSH	279 ± 20		
		Middle cerebral artery occlusion (MCAO):			
		Reduced GSH	593 ± 35		
		Oxidized GSH	204 ± 22		
	Blood	140 Individuals (33 female, 37 male):			[93]
		Control group (volunteers with similar cerebrovascular risk factors), N = 70	2.3 ± 0.4		
		Patients with acute ischemic stroke, N = 70	3.9 ± 2.5		
S-adenosylhomocysteine methylation (%)	Whole peripheral blood	202 individuals:		In patients with ischemic stroke, the percentage of methylated reference AHCY was significantly higher than in controls.	[94]
		Patients with acute ischemic stroke, N = 64	0.13% (0.09%, 0.27%)		
		Control group, N = 138	0.06% (0.00%, 0.17%)		
Methionine (odds ratios (oRs)	Plasma	PREDIMED Cohort, 567 women: Stroke cases, N = 59 Controls, N = 508	OR 1.85 (95% CI 1.44–2.37)	Methionine sulfoxide was linked to an increased risk of stroke	[95]

**Table 2 ijms-24-06389-t002:** Detection strategy of oxidative stress biomarkers.

Oxidative Stress Biomarkers	Detection Strategy Examples	References
Malondialdehyde	UV-Vis Fluorescence Electrochemistry GC-MS SERS TBARS Test	[96,97]
F2-isoprostanes	GC-MS GC-NICI-MS ELISA HPLC-MS/MS SPE-HPLC-MS/MS	[98,99,100,101]
4-Hydroxy-2-Nonenal	2-AP HPLC Sandwich ELISA Western Blot LC-MSMS FT-ICR MS MALDI-TOF-MS ^32^P-PostlabelingGC-MS DNPH Derivatization	[102,103,104]
Oxidized LDL	TBARS assay ELISA	[105,106]
8-oxo-7,8-dihydro-2-deoxyguanosine	HPLC-ED	[107]
3-Nitrotyrosine	(HPLC)-(UV-VIS) absorption Electrochemical (ECD) Diode array (DAD) LC-MS LC-MS/MS GC-MS GC-MS/MS Sandwich ELISA	[108,109]
Gluthatione	DTNB/GR enzyme recycling method HPLC	[110]
Protein carbonyls	Western blot In-gel fluorophoric tagging Levine spectrophotometric method ELISA HPLC	[111,112]
Homocysteine	HPLC with fluorometric detection HPLC-ED Immunonephelometric method LC-MS-MS Fluorescence polarization Immunoassay EIA	[113]
Methionine sulfoxide	Peptide mapping with MS detection rpHPLC HIC Weak cation-exchange chromatography	[114]
Myeloperoxidase	BLI CRET ADHP MPO-Gd MR imaging	[115,116]

GC-MS: gas chromatography-mass spectrometry; NICI-MS: gas chromatography-negative-ion chemical ionization mass spectrometry; HPLC: high performance liquid chromatography; SPE: polymeric weak anion-exchange solid-phase extraction; UHPLC–MS/MS: isotope-dilution ultrahigh performance liquid chromatography electrospray ionization–tandem mass spectrometry; 2-AP: fluorescent probe 2-aminopyridine; DNPH: 2,4-dinitrophenylhydrazine; HPLC-ED: high performance liquid chromatography with electrochemical detection; EIA: enzyme-linked immunoassay; BLI: bioluminescence imaging; CRET: chemiluminescence resonance energy transfer; MPO-Gd: bis-5-hydroxytryptamide-diethylenetriaminepentaacetate-gadolinium; ADHP: 10-acetyl-3,7-dihydroxyphenoxazine; TBARS: thiobarbituric Acid Reactive Substances.

## 3. DNA Oxidation

Commonly and in pathological conditions, DNA oxidation leads to the breakage of DNA strands, loss of bases, nucleotide oxidation, and adducts formation [117]. Under normal circumstances, DNA oxidation occurs in learning and memory formation and is due to ROS actions against guanine [118]; guanine’s repair is carried out via base excision repair pathway [119]. However, when excess ROS is accumulating in cells, they can attack DNA molecules, and if the repair mechanisms cannot overcome the extent of the oxidative stress-driven changes to DNA molecules, the oxidation products can be found in some biological fluids of the affected individual (blood, urine, saliva) [120,121]. If the oxidized DNA product, 8-oxo-7,8-dihydro-2-deoxyguanosine (8-oHdG), is not successfully removed by the enzymatic repair pathway, it will participate in epigenetic alterations, mutagenesis, and altered gene regulation [122]. Mitochondrial DNA is much more susceptible to oxidative stress compared to nuclear DNA [123]. Hence, 8-oHdG could be a sensitive biomarker for oxidative stress-driven DNA damage of the brain mitochondrial DNA [86].

Similar to other pathological conditions characterized by multifactorial interaction between oxidative stress and inflammation, 8-oHdG is a highly specific and abundantly produced compound during oxidative damage [56]. In ischemic stroke, severe oxidative damage results in relatively high concentrations of 8-oHdG in blood serum and urine of patients compared to healthy individuals [87,88]. In addition, Hsieh et al. [124] reported that 8-oHdG can be used as a reliable biomarker of oxidative stress to monitor the functional outcomes of stroke patients after rehabilitation. A cohort study reported that the mean urinary level of 8-oHdG on Day 7 was significantly high in 44 stroke patients compared to controls. They further reported that the higher 8-oHdG levels were proportional to poor outcomes in stroke patients [88] (Table 1). Similarly, Liu et al. [63] reported an association between urinary levels of 8-oHdG and MDA in predicting post-stroke cognitive impairment after 1 month.

## 4. Oxidative Protein Modifications

Relatively few studies have reported on the usefulness of proteins’ oxidation products as biomarkers in acute ischemic stroke. Protein-consisting amino acids oxidation often results in reversible and non-reversible effects on the affected protein functions and structure [124]. When exposed to ROS, highly reactive sulfur-containing amino acids (cysteine, methionine) could form cross-linkages and alter the proteins’ physical as well as chemical properties [125,126]. The association of common oxidation products of proteins with ischemic stroke is discussed in this section.

### 4.1. Protein Carbonyls

In the presence of metal ions, several amino acids within protein structures are oxidized by ROS and other reactive species [127]. Therefore, protein carbonyls are formed. In some cases, a second wave of reaction with ROS or products could lead to aldehyde addition [128]. In other cases, metal-modulated non-enzymatic catalysis is not necessary, while in different circumstances, lipid peroxidation products of some amino acids undergo oxidation [127].

Regarding the potential of protein carbonyls as ischemic stroke biomarkers, the scientific databases show a severe lack of evidence. Žitňanová et al. [48] reported a strong negative correlation between levels of plasma protein carbonyls and total antioxidant capacity (TEAC) in the control groups (Table 1), while there was no significant change in the plasma protein carbonyl levels of patients who suffered acute ischemic stroke. In a recently published study, Pawluk et al. [21] reported a positive correlation between levels of protein carbonyls and functional outcomes after acute ischemic stroke, suggesting that carbonyl group determination could only be a potential marker of protein damage following cerebral thrombolytic treatment.

### 4.2. 3-Nitrotyrosine

3-nitrotyrosine is a versatile oxidative stress marker that generally predicts oxidative and nitrosative damage to proteins. During the inflammatory response, the macrophages could easily produce significant amounts of superoxide and nitric oxide that further participate in peroxynitrite synthesis that contributes to tyrosine residue nitration [129]. As a consequence of oxidative and nitrosative stress, considerable amounts of 3-nitrotyrosine are synthesized and are then present in the blood, cerebrospinal fluid, bronchoalveolar fluid, semen, and urine [130].

Among the few recent studies regarding the use of 3-nitrotyrosine as biomarker in acute ischemic stroke, Medeiros et al. [131] showed a putative association between levels of 3-nitrotyrosine and ischemic stroke. Since fibrinogen is an abundant plasma protein, the authors hypothesized that nitration of fibrinogen molecules by free radicals may occur very early during ischemic stroke events, and therefore can also be used as a potential biomarker for its diagnosis. In addition, they also reported the sensitivity of three out of nine tyrosine residues in conditions of ischemic stroke compared to non-inflammatory diseases. They further demonstrated different tyrosine nitration patterns through a nitration curve suggesting the sensitivity of mapping fibrinogen modification in ischemic stroke. Cichon et al. [132] reported a correlation between the oxidative damage of proteins and degree of post-stroke depression.

### 4.3. Ischemia-Modified Albumin (IMA)

Generally, ischemia modified albumin (IMA) is a newly recognized sensitive biomarker of ischemic diseases [133]. During ischemic stress, the circulating albumin is modified due to oxidative stress, and pH decreases. In this way, the N-terminal end of the albumin molecule decreases in reactivity to metal compounds (cobalt, copper, and nickel) due to auto-degradation or the damaging effects of the theoretical Fenton reaction products [134]. In stroke patients, serum IMA levels were reported to be higher during the acute phase of ischemic stroke, and then gradually declined over 1 week [135]. Hence, the authors suggested that IMA could be a rapid biomarker for the prediction of early ischemic stroke. In another study, Jena et al. [136] showed the correlation between acute stroke and IMA levels in 33 hemorrhagic and 35 thrombotic stroke patients. They determined the serum IMA and MDA levels in the test patients and reported high sensitivity of IMA compared to MDA in predicting the incidence of strokes. Similarly, Okda et al. [137] reported that IMA levels could significantly differentiate between ischemic stroke and hemorrhagic stroke. The study also compared IMA levels with standard scoring systems, such as the National Institute of Health Stroke Scale (NIHSS) and Glasgow Coma Scale (GCS) and found a positive correlation with the former and an inverse one with the latter.

### 4.4. Advanced Glycation End Products

Advanced glycation end products (AGEs) are a group of compounds with complex structure and reactivity [138] that originate from the interaction of lipids or proteins with sugars via glycation [138]. Their relevance in the prognosis of diabetes mellitus, chronic kidney diseases, and cardiovascular diseases is well established [138,139,140]. However, very few studies have reported its role in relation to cerebral ischemia. Selective AGEs have been investigated as biomarkers, and high levels of serum pentosidine have been associated with the prediction of poor outcome after acute ischemic stroke [141]. Similar findings were also reported by Yokota [142].

Compared to AGEs, the receptor for AGEs (RAGE) and its isoform (sRAGE) have been more commonly used in recent years as a biomarker for ischemic stroke. RAGE is a multi-ligand protein, and its expression itself is linked to oxidative stress induction, whereas sRAGE exerts cytoprotective effects by competing with RAGE for ligands. Park et al. [143] associated low plasma sRAGE levels with high acute ischemic stroke risk and severity. An interesting study by Montaner et al. [144] reported that combined determination of sRAGE along with five other biomarkers, including D-Dimer, chimerin, secretagogin, caspa-se-3, and MMP-9, increases the predictive probability of acute stroke to 99.01%.

### 4.5. Advanced Lipoxigenation End Products

Advanced lipoxigenation end products (ALEs) are formed on non-enzymatic chemical modification of reactive carbonyl species that are produced during membrane phospholipids acyl chains peroxidation [145]. ALEs are highly stable compounds and directly relate to cellular damage and dysfunction [146]. However, their use as molecular biomarkers is severely limited in acute ischemic stroke. One of the main reasons for their limited use is the non-sensitivity and non-specificity in most diseases. ALEs are primarily associated with the oxidation of LDLs and the formation of atherosclerosis lesions, and thus they could only signify a non-specific estimate of stenosis predisposing disorders, such as diabetes, cardiovascular diseases, and chronic kidney diseases [147,148].

### 4.6. Methionine Sulfoxide

Methionine sulfoxide can also be considered an in vivo biomarker of oxidative stress. It is a major derivative of methionine oxidation with ROS by a two-electron dependent mechanism [149]. Balasubramanian et al. [95] reported an association between the methionine sulfoxide and the increased risk of incident stroke. Another study conducted by Li et al. [150] observed a neuroprotective impact in ischemia/reperfusion injuries when when an increase in methionine sulfoxide reductases enzymes occurred following betaine administration; the methionine sulfoxide reductases enzymes. As methionine oxidation was found to contribute to cerebral ischemia/reperfusion injury through the potentialization of NF-κB–dependent adhesion molecule activation [151], methionine sulfoxide might be considered a potential marker of oxidative stress in acute ischemic stroke.

### 4.7. Homocysteine

Homocysteine (2-amino-4-mercaptobutyric acid) is a non-proteinogenic sulfur-containing amino acid derived from demethylation of methionine via S-adenosylmethionine (SAM) and S-adenosylhomocysteine (SAH) [152]. Homocysteine metabolism represents an intersection of two pathways: (1) remethylation pathway regenerating methionine via methylfolate homocysteine methyltransferase (methionine synthase), its coenzyme, methylcobalamin, and 5-methyltetrahydrofolate (MTR) or betaine-homocysteine methyltransferase (BHMT) as the methyl donor; (2) transsulfuration pathway generating cystathionine via cystathionine β-synthase (CBS), the cystathionine thus formed is then transformed into cysteine via cystathionine γ-lyase (CTH); both enzymes require the cofactor pyridoxal phosphate (vitamin B6) [152,153]. Both prospective and retrospective investigations have revealed a link between plasma homocysteine levels and the risk of ischemic stroke [89,90,154,155]. Epidemiological data suggest that homocysteine is a significant predictor of ischemic stroke. Moderately elevated homocysteine values (even in the population reference range) are linked to vascular pathology by a variety of mechanisms, including atherosclerotic and thrombotic events [154]. Patients with ischemic stroke, both in the acute and convalescent phases, have been found to have hyperhomocysteinemia [156,157]. A nested case-control study from the Netherlands showed that a 1 µmol/L increase in total homocysteine concentrations was associated with an increase in stroke risk from 6% to 7%. The study also showed an increased risk of stroke above the highest quintile (18.6 µmol/L) compared with the lowest quintile (12.0 µmol/L), with an odds ratio of 2.43 (1.11 to 5.35) [89]. Similarly, Perry et al. [90] found that, among middle-aged British men, stroke patients had considerably higher serum total homocysteine concentrations (across quartiles) (geometric mean 11.9 [11.3–12.6] µmol/L) than in controls (geometric mean 13.7 [12.7–14.8] mol/L). In comparison to the first quarter, there was a gradual quartile rise in the relative stroke risk in the second, third, and fourth quarters of the total homocysteine distribution (odds ratios 1–3, 1–9, 28; trend *p* = 0–0.05) [90]. Consistent with prior research in older people, Niazi et al. [91] showed that 50.7% of young ischemic stroke patients exhibited a moderate to high frequency of homocysteine. They also found that homocysteine levels were notably higher in men (mainly in the 36–45 age group) than in women [91]. Perry et al. [90], Bots et al. [89], and Niazi et al. [91] characterized hyperhomocysteinemia (hHcy) as values > 12μmol/L. These three investigations reported a significant correlation between hHcy and stroke incidence, implying that hHcy can be used as a predictive risk factor for stroke progression [89,90,91]. Furthermore, Zhao et al. assessed the ability of S-adénosylhomocystéine hydrolase (AHCY), an enzyme responsible for catalysis of S-adenosylhomocysteine into adenosine and homocysteine, for predicting the outcomes of IS [94]. The study showed that the percentage of DNA methylation of AHCY was significantly (*p* < 0.0001) higher in patients with ischemic stroke 0.13% (0.09%, 0.27%) than in controls 0.06% (0.00%, 0.17%) [94].

### 4.8. Glutathione

Glutathione (γ-L-Glutamyl-L-cysteinylglycine) is a low molecular weight sulfur-containing pseudo-tripeptide formed by the condensation of glutamic acid, cysteine, and glycine. It is the major non-protein thiol in many tissues. It is involved in various vital processes, including redox homeostatic buffering [158]. It participates in the protection of protein thiol groups against oxidative damage and the neutralization of reactive oxygen species. Glutathione exists in two forms: the reduced form (GSH) and the oxidized form (GSSG). Over 90% of total glutathione (GSH) is in the reduced form (GSH). In the brain, glutathione is mainly in a reduced form in high concentration (~1–3 mM) [158,159,160]. In two experimental rat models of ischemia caused by hypoperfusion (middle cerebral artery occlusion and bilateral occlusion of the common carotid arteries), GSH homeostasis as an oxidative stress marker has been disturbed in global and focal ischemia. A drop in reduced and a significant increase in oxidized GSH was shown in the brain on rat models of cerebral ischemia after MCAO and BCAO. This effect may be attributed to the fact that the dynamic GSH homeostasis shifts toward the oxidized form (GSSG) as a result of oxidation [92]. In human study, Ozkul et al. [93] reported an increase in serum GSH levels in 70 subjects within 48 h after stroke compared with matched controls [93]. Similar to this, Zimmermann et al. [161] noted an increase in GSH and GPX levels during the first 6 h and 1 day after the acute stroke, respectively, when compared to controls [161]. According to these results, the increased GSH levels could be part of first-line defense mechanisms against oxidative stress and could provide adaptive mechanisms to oxidative stress during AIS [93,161].

## 5. Changes in Protein Coding Gene

### Myeloperoxidase

Myeloperoxidase is a heme-containing peroxidase enzyme expressed in various cells including neurons [162]. Early studies have identified the increased risk of ischemic stroke with genetic variability of myeloperoxidase [163,164,165]. It is also associated with the severity of brain damage and functional outcomes after stroke. Many studies have reported the myeloperoxidase polymorphisms and their consequently increased activity in ischemic stroke patients [166,167,168] and that the inflammatory response mediated by myeloperoxidase may be an underlying mechanism of brain injury in ischemic stroke [169,170]. The definite role of myeloperoxidase activation in oxidative damage and ischemic stroke was confirmed by Forghani et al. [171] and Kim et al. [172] who studied the effect of myeloperoxidase inhibitor on infarct size and volume in rodent ischemic stroke models. Both studies reported that the administration of myeloperoxidase inhibitors leads to decreased inflammation and reduced infarct size and volume. In addition, the neurological deficits were significantly low and survival rates were very high compared to controls. More specifically, Liu et al. [173] reported the rs2107545 polymorphism of the myeloperoxidase gene as a sensitive biomarker for the prediction and prognosis of ischemic stroke. The findings of this study were validated based on observations of 351 ischemic stroke patients and 417 controls.

The myeloperoxidase (MPO) catalytic cycle. Ferric myeloperoxidase (MPO-Fe3+ in its resting state) reacts reversibly with hydrogen peroxide by oxidation of the heme group and formation of the ferryl-p cation radical intermediate compound I are also described in the Figure 3.

Despite the important role of oxidative stress in the pathogenesis of ischemic stroke, diagnosing ischemic stroke using oxidative stress markers remains a significant challenge since oxidative stress occur in many diseases associated with energy metabolism impairment, neuroinflammation, tissue damage, and cell loss. However, the relevance of oxidative stress markers could still be endorsed in monitoring the ischemic stroke severity and prognosis.

## 6. Measuring the Modification of Antioxidant Defense Status as Potential Indirect Markers of Oxidative Stress

Endogenous antioxidants are the primary cellular defense in counteracting the negative impact of free radicals, such as ROS. In fact, under normal physiological conditions, there are three distinct antioxidant systems: enzymatic antioxidant defense, non-enzymatic antioxidant defense, and repair enzymes [174]. The evaluation of each category activity is thought to have different relevance in many different pathological conditions that involve the deregulation of oxidative homeostasis. As ROS are extremely unstable and highly reactive, they quickly react with cellular constituents or are catabolized, true evaluation of the ROS content is not possible. Oxidative homeostasis status can, therefore, only be indirectly determined by assessing oxidative stress products [175], such as the ones previously discussed in the current context. However, the evaluation of the latter could not provide information about the oxidative homeostasis mechanisms that are impaired, except for the general meaning of ROS overproduction and accumulation. In order to find relevant information on the altered mechanisms, the actual participants in the oxidant/antioxidant gearing need to be evaluated. Consequently, numerous analytical techniques have been developed to enable the antioxidant defense evaluation as a reliable yet indirect biomarker of oxidative stress.

### 6.1. Antioxidant Enzymes

The antioxidant enzymes are mainly responsible for ROS enzymatic degradation. In ischemic stroke, the evaluation of antioxidant enzymes was extensively carried out: superoxide dismutase (SOD), glutathione peroxidase (GPx), and catalase (CAT). Their main role is to perform ROS gradual degradation or to neutralize reactive potential. The extensive description of the antioxidant enzymes’ role was previously carried out in many studies [176,177,178,179]; briefly, SOD is responsible for the degradation of superoxide ions by reducing them to hydrogen peroxyl, which is a less harmful ROS, with or without the help of metallic ions (copper, zinc, manganese), while GPx and CAT are actively implicated in the latter neutralization to oxygen and water [176].

As many of the studies regarding the antioxidant enzyme activities in different pathologies often evaluate more than one antioxidant enzyme, their evaluation could be found in a multi-panel-like assessment in AIS. Animal model studies showed that ischemic stroke is generally followed by enzymatic antioxidant defense weakening, with dramatic falls in SOD, GPx, and CAT activities [180]. In acute ischemic stroke patients, SOD and CAT activities were found to be significantly decreased at 1 day following the acute ischemic stroke, and their activity tended to normalize as recovery was achieved, while GPx followed an inverse activity dynamic [181]. Changes in the antioxidant enzymes gene expression (SOD, GPx, and CAT) were also reported [182]. However, Valavi et al. [183] failed to find any differences in antioxidant enzymes activity (SOD, GPx, CAT) in acute ischemic stroke patients, though a similar trend was observed in hemorrhagic stroke patients for SOD activity (day 1 versus day 7). As they are non-specific biomarkers that generally point out oxidative stress development rather than a specific pathological condition, and considering the controversial results of the recent studies, it could be challenging to find possible use of antioxidant enzymes activities assessment in acute ischemic stroke diagnosis. Thus, further studies are needed to address this limitation. On the other hand, a recent study showed that there are significant enzymatic activity dynamics between the two main stroke types and in direct correlation with NIHSS, thus gaining possible predictive value [184].

### 6.2. Non-Enzymatic Antioxidants

By contrast to enzymatic antioxidants that are directly implicated in ROS metabolism, the non-enzymatic antioxidants are mainly ROS scavengers. Alongside the endogenous enzymatic antioxidants, the non-enzymatic antioxidants mainly originating from dietary sources (fruits and vegetables) contribute to oxidative homeostasis [185]. The yielding in ROS scavenging of the non-enzymatic antioxidants that may be vitamins (C, E), enzymatic cofactors (coenzyme Q10), or other cellular constituents, could be measured by assessing several antioxidant indices, such as total antioxidant power, total and perchloric acid oxygen radical absorbance capacity, or total peroxyl radical–trapping potential.

### 6.3. Total Antioxidant Power

Total Antioxidant Power (TAP) could evaluate the overall status of oxidative homeostasis by measuring the free radical-reducing capacity of all dietary non-enzymatic antioxidants (hydrophilic and lipophilic). Additionally, TAP could provide information about the synergistic effects between different antioxidants [49,181,186,187].

To date, there is scarce data on the TAP evaluation in AIS by relation to incidence, severity, or outcomes. Moreover, the few available studies have provided contradictory observations with respect to TAP levels in AIS patients. For an instance, there are some studies that associated high TAP levels with the AIS incidence [49,186], whereas other studies have reported lower TAP levels for the same pathology [181,187], or no significant difference between the circulating TAP in AIS compared to controls [45,188]. Though the high TAP levels reported in acute ischemic stroke are seemingly creating confusion, Lorente et al. [186] provided a plausible explanation by including the surviving and non-surviving ischemic stroke patient’s data in the statistical analysis. Thus, the higher-than-average TAP levels in surviving acute ischemic stroke patients indicated high individual resilience to oxidative state, probably due to individual immune factors variability. Meanwhile, it was suggested that the non-survivors had lower resilience for oxidative stress and higher circulating TAP levels and responded to acute ischemic stroke in a ‘fight till death’ manner. Consequently, a new valuable mechanistic detail on the oxidative stress implication in acute ischemic stroke emerged which may explain the survival strategy of acute ischemic stroke patients: resilience to oxidative stress and non-excessive immune system activation.

However, more studies are needed to understand the association between TAP and the degree of inflammation during oxidative stress at the mechanism level. Based on the outcomes of 41,620 patients and highlighting the role and mechanism of different dietary nutrients, Rio et al. [189] reported that antioxidants play a significant role in reducing oxidative stress. They may also play a role in reducing the risk of cerebral infarction. Despite this promising evidence, caution should be advised in antioxidant therapy, as Rio et al. also showed that high intake of vitamin E positively correlates with the risk of hemorrhagic stroke. 

### 6.4. Total and Perchloric Acid Oxygen Radical Absorbance Capacity

The ORAC_PCA_ assay provides a direct measure of antioxidant capacity of the serum nonprotein fraction treated with perchloric acid (PCA) [190]. However, when ORACTOT, a biomarker of plasma, is precipitated with plasma proteins that have antioxidant capacity, by 0.5 M perchloric acid (ORACPCA) in a 1:1 ratio, the sample reflects the antioxidant capacity of the remaining small-molecular-weight compounds (ORAC_TOT_ values are generally lower than ORAC_PCA_) [46]. While the sample is considered very sensitive because it is based on the use of the most prevalent free radical in humans (peroxyl radicals), its flexibility for using free radicals [46,191] is well known. 

Lorenzano et al. [46] reported the reliability of ORAC and F2-isoprostane in predicting the outcome during ischemic strokes. Specifically, they reported that ORAC_PCA_ and total oxygen radical absorbance capacity (ORAC_TOT_) values independently correlated with ‘mismatch’ and ‘mismatch salvage’ in their assessment of diffusion-perfusion mismatch in AIS patients. Here ‘mismatch’ and ‘mismatch salvage’ referred to the overlapping and non-overlapping regions of cerebral infarction observed by brain imaging techniques.

### 6.5. Total Peroxyl Radical–Trapping Potential

Total peroxyl radical–trapping potential (TRAP) is another useful parameter for measuring total antioxidant activity, but its specificity for a particular disease is reduced [192], including its potential to predict or diagnose acute ischemic stroke. The general significance of this evaluation is to shed more light onto the capacity to resist the ROS attack against the cells [193] due to the combined activity of all cellular antioxidants [192].

Among the few published studies that used TRAP to measure antioxidant potential in cerebral infarction, Leinonen et al. [194] reported that TRAP exhibits significant inverse correlation with NIHSS and Hospital Mobility Scale (HMS) and direct correlation with the Barthel index score (BIS) in patients with cerebral infarction, suggesting that further studies could propose TRAP as a potential biomarker for disease severity, or disease outcomes, or be useful during rehabilitation.

## 7. Future Treatment Directions

Even though the proper use of oxidative stress biomarkers in acute ischemic stroke diagnosis and prognosis has not yet been fully elucidated, substantial evidence shows that oxidative stress is an important acute ischemic stroke progression factor. Oxidative stress therefore becomes the most relevant target in acute ischemic stroke treatment and prevention.

### 7.1. Targeting Oxidative Stress May Be a Potential Treatment for Ischemic Stroke

Many therapeutic approaches targeting the ROS species have been explored for the treatment of acute ischemic stroke [195] being described as ‘upstream’ (attenuation of ROS production) or ‘downstream’ (ROS neutralization) strategies. While studies using in vivo models of acute ischemic stroke have suggested the positive effects of antioxidants [196,197,198], the clinical trials have not shown promising outcomes, probably due to the lack of understanding regarding the contributing factors that influence ischemic stroke progression, including ROS [38,68,69,70,71]. A detailed review on the oxidative stress-targeted therapeutic options in ischemic stroke treatment [38] suggested that the most practical downstream approaches include ROS scavengers (vitamin C, D, E, NAC, NXY-059, edaravone, tirilazad [U-74006F], citicoline, and lipoic acid) and ROS degraders (such as polyethylene glycol-conjugated superoxide dismutase). Another a possible explanation for ROS-targeted treatments failure in ischemic stroke was proposed because ROS production could be easier to prevent, which the scavenging, degrading, and neutralizing approach is more difficult. In this context, targeting upstream strategies, such as the inhibition of NOX, XO, cyclooxygenase-2, MTA, and CoQ10, may be more effective. In addition to these factors, Wu et al. [199] suggested that, since the interaction of oxidative stress and inflammation aggravate cerebral infarction, the inhibition of the factors contributing to the modulation of both of these processes (such as superoxide anion free radicals, hydroxyl free radicals, and nitric oxide) could be a potential strategy for ischemic stroke treatment. Wang et al. [200] evaluated various pathological processes in ischemic stroke and reported the critical role of nuclear factor erythroid 2-related factor 2 (Nrf2) in signaling pathways such as Keap1, PI3K/AKT, MAPK, NF-κB, and HO-1, which alleviate oxidative stress during ischemia, meaning that targeting nuclear factor erythroid 2-related factor 2 (Nrf2) can be an effective strategy in inhibiting oxidative stress. In another study, an indirect approach of preconditioning hypoxia-inducible factors with prolyl-4-hydroxylase domain enzyme inhibitors showed neuroprotective effects by modulation of synaptic plasticity in mice models [201].

### 7.2. Antioxidant Can Be Explored for Their Potential to Prevent Ischemic Stroke 

The critical role of antioxidants in improving overall immunity against infectious diseases is well-known [202]. Similarly, the antioxidant therapy has also shown promising results in the management of chronic non-communicable diseases [203]. However, its preventive properties have not been fully described to date [203,204,205]. Multiple factors need to be addressed while exploring the preventive effects of antioxidants in ischemic stroke and other diseases, including lifestyle, comorbidities, and possible sources of antioxidants, age, and environmental pollution [204]. It is also important to note that the high antioxidant potential originating from a plant source does not imply its complete bioavailability to humans [199]. On the other hand, if antioxidant solutions are chemically synthesized, long-term administration could be cautiously recommended due to their potential harmful effects [204]. Consequently, while it could be challenging to prevent these oxidative stress-aggravated diseases in genetically predisposed individuals, it might be achievable in low-risk individuals (risk originates from environmental and lifestyle reasons) [205,206].

Antioxidants from dietary sources, including fruits and vegetables, may be the key factor in preventing ischemic stroke and related diseases. Alternative and complementary medicine could also offer promising adjuvant therapy based on the medicinal plants that provide antioxidant activity. In a recent review on the alternatives and solutions that herbal medicine could offer for ischemic stroke, Gaire et al. [207] described more than 30 medicinal plant species that could be useful, many of which are known for their antioxidant effects because of their increased polyphenols, flavanols, quinones, lignans, and sphingolipids content. Examples include common plant species, such as *Curcuma longa*, *Panax ginseng*, *Camellia sinensis*, or *Ginkgo biloba*, that were previously tested in ischemic stroke conditions, and more specific plants, such as *Crataegus oxyacantha*, *Cordyceps sinensis*, *Paeonia suffruticosa*, or *Scutellaria baicalensis*. These were all shown to exhibit positive effects in acute ischemic stroke animal models by mostly interacting with the oxidative stress pathways [207]. Similarly, *Medicago sativa* extract was found to improve antioxidant enzymes (SOD and GPx) and anti-inflammatory cytokines (TNFa and IL1) activities due to its increased content of phenols, flavonoids, isoflavones, phytoestrogens, and vitamin C [207,208,209].

The antioxidant potential of many foods and condiments were also shown in pathological conditions (for example, the recent studies on the antioxidant potential of *Brassica oleracea var. italica* buds extract (rich in sulphoraphane and phenols) and the *Origanum vulgare* leaves extract (rich in phenols and flavonoids) [210,211,212,213]. However, the increased use of chemical fertilizers may alter the properties of phytonutrients and bioactive compounds leading to unwanted negative effects. Hence, the intake of naturally grown chemical-free products that additionally exhibit antioxidant properties could be a potential preventive approach. Although, there is no relevant evidence to support the potential of antioxidants from organic sources in preventing or managing the consequences of ischemic stroke, abundant evidence is available about the negative effects of the excessive use of chemicals in growing foods that leads to altered plant metabolism and health problems for consumers [213,214].

## 8. Conclusions

Ischemic strokes often occur due to atherogenic or thrombogenic events leading to arterial obstruction. As a main consequence, the blood supply to areas of the brain tissues could be reduced or completely interrupted, leading to tissue damage and cellular apoptosis. Even though compelling evidence on the implication of oxidative stress in the development of the ischemic process is still needed, it is currently accepted that the reactive oxygen species accumulation resulting from ischemia-related oxidative dyshomeostasis encourages oxidative stress-associated molecular and cellular damage. Accordingly, the assessment of oxidative stress biomarkers could be useful in describing the pathogenic mechanisms of ischemic strokes, as well as providing promising diagnosis and prognosis tools and offering guidance to help find new targets for antioxidant-based therapy. However, even though some of the current diagnosis measures in ischemic stroke are imperfect, further studies are needed to overcome the limitations of the potential oxidative stress biomarkers to predict, diagnose, and follow up on the efficiency of treatments, mainly due their nonspecific character. At the same time, while several relevant studies acknowledge the potential of oxidative homeostasis modulation in ischemic stroke prevention and management, the lack of clinical studies and their controversial results avert agreeable conclusions.

## Figures and Tables

**Figure 1 ijms-24-06389-f001:**
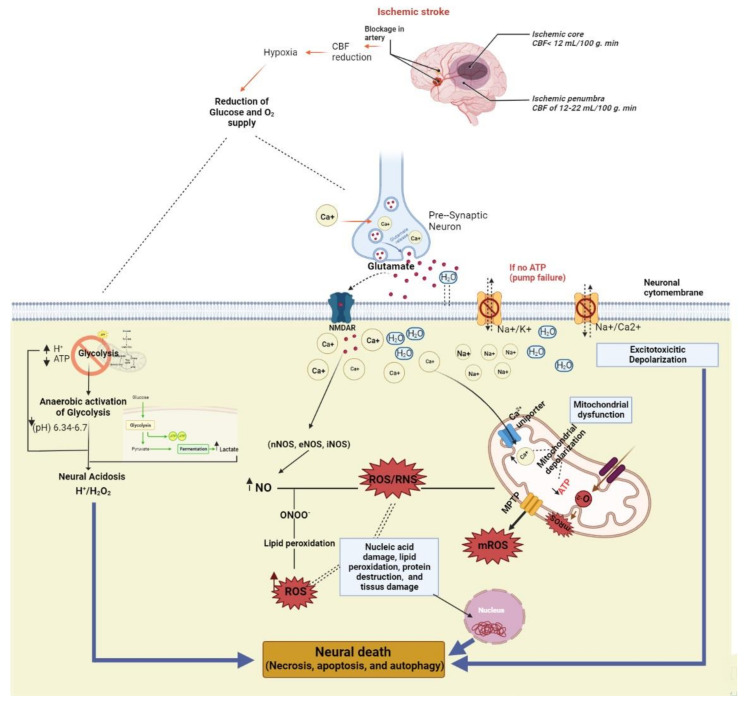
Oxidative stress pathways in pathophysiology of ischemic strokes.

**Figure 2 ijms-24-06389-f002:**
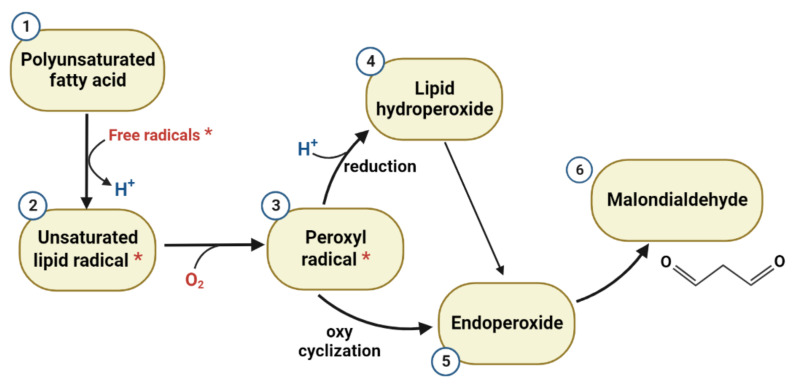
MDA generation cycle: (1) the high reactivity of polyunsaturated fatty acids to oxidizing agents is promoted by the methylene groups positioned between the cis double bonds along the chain. The removal of the hydrogen atoms leads to unsaturated lipid carbon-centered radicals (2) that are vulnerable to oxidizing and conversion into highly unstable peroxyl radicals (3). Depending on the position of the peroxyl radical in the carbon chain, two pathways are possible: terminal position at one of the two ends of the double bond system will result in reduction to conjugated diene hydroperoxides (4), while internal position will result in multiple products by mono- and/or bi-cyclizations, coupling to O2 or reductions. The fragmentation of a bicyclic peroxide compound (5) structurally analogous to the prostaglandin endoperoxide generates the malondialdehyde (6), *: free electron.

**Figure 3 ijms-24-06389-f003:**
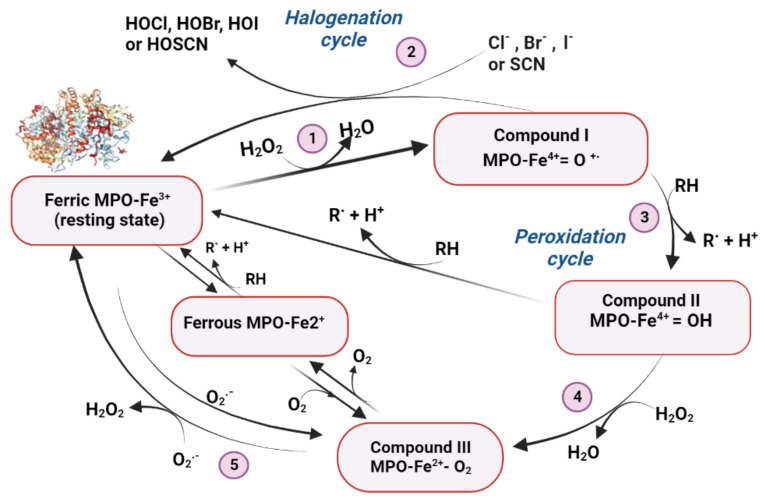
The myeloperoxidase (MPO) catalytic cycle. Ferric myeloperoxidase (MPO-Fe3+ in its resting state) reacts reversibly with hydrogen peroxide by oxidation of the heme group and formation of the ferryl-p cation radical intermediate compound I (1). This intermediate can oxidize halides and the pseudo-halide thiocyanate to the corresponding hypohalous acids, while being readily reduced to MPO-Fe_3_+ (2). Compound I also oxidizes multiple organic substrates (RH) to free radicals (R.) in the classical peroxidase cycle which involves two sequential one-electron reduction steps, generating compounds II (3) and III (4). Peroxidation substrates include tyrosine estradiol, serotonin, norepinephrine, ascorbate, urate, etc. Superoxide can also be the substrate for compound I and compound II in these stages. Ferric myeloperoxidase can be reduced to its ferrous form MPO-Fe_2_+, which binds to O_2_, forming the oxymyeloperoxidase, or compound III (MPO-Fe_2_+O_2_). Superoxide also provides the substrate for the fast conversion of the ferric enzyme to form compound III (5). Ferrous myeloperoxidase and compound III can display catalytic action by reductively activating quinones (adapted from Podrez et al., [173]).

## Data Availability

All data is available on request.

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
