# Peer review of "The Role of Potential Oxidative Biomarkers in the Prognosis of Acute Ischemic Stroke and the Exploration of Antioxidants as Possible Preventive and Treatment Options"

_ijms, 2023, doi:10.3390/ijms24076389_

Round 1

Reviewer 1 Report

In this review, the authors summarize the current knowledge regarding the role of oxidative stress markers on the prognosis and monitoring of ischemic stroke. They have also tried to collect evidence about the efforts that have taken place in order to therapeutically target oxidative stress. Before diving into the role of oxidative stress as biomarker, the authors have tried to explain the molecular mechanisms provoking the formation of ROS, and the mediators of oxidative stress-induced cellular damage. I think that they need to better describe these mechanisms.

In general, the presentation of the studies that evaluated the strength of oxidative stress biomarkes in ischemic stroke is sufficient and vey well-done. The references cited are up-to-date, but also include seminal studies that are not so recent. The authors have already published original research articles related to oxidative stress and the role of antioxidants in disease prevention in high-impact journals. Hence, their expertise is considered sufficient. However, the authors state that this is a synthetic review. I would consider it as a narrative review.

ABSTRACT

Line 25: severity-reflecting instead of severity-dependent?

Line 31: before “thus” end the previous sentence. I think it is too long already.

Lines 35-37: this sentence seems to me like a repetition of the previous one.

INTRODUCTION

Page 1, line 45: renders them instead of makes them?

Page 2, line 47: pathogenic mechanisms instead of pathological?

Line 54: research instead of re-search

Line 61: risk factors for the ischemic stroke

Line 67: being instead of are

Lines 114-117: I do not understand what you are trying to express. Do you mean that energy depletion precedes oxidative stress?

General comment: please be more precise about the molecular mechanisms leading to ROS formation in stroke.

Lines 128-132: how did you conclude that the existence of cell-free mitochondrial DNA is a marker of increased ROS? Circulating mitochondrial DNA is increased due to tissue damage in general.

You seem to use the words diagnostic and prognostic instead of diagnosis and prognosis

Figure 1: some parts of the figure are not readable. Why do you say “damage cytoskeleton” and then in the brackets you write all the  types of macromolecules?

Line 179: the definition of ferroptosis is by too simplistic.

Table 1 is very well structured and provides really meaningful information

Why did the authors selectively discuss about myeloperoxidase?

Please, use shorter sentences. It has been really hard to follow.

I believe the authors should emphasize that oxidative stress biomarkers can be used mainly on monitoring the severity and prognosis. Their role in diagnosis is unlikely; increased ROS are implicated in many diseases. According to this review, it does not seem to exist a marker to discriminate among these diseases.

Author Response

Reviewer 1

“In this review, the authors summarize the current knowledge regarding the role of oxidative stress markers on the prognosis and monitoring of ischemic stroke. They have also tried to collect evidence about the efforts that have taken place in order to therapeutically target oxidative stress. Before diving into the role of oxidative stress as biomarker, the authors have tried to explain the molecular mechanisms provoking the formation of ROS, and the mediators of oxidative stress-induced cellular damage. I think that they need to better describe these mechanisms.”

Response: We would like to thank the Reviewer for the comments and for classification of the study as sufficient and very well-done. We have carefully read the comments and have revised/ completed the manuscript accordingly. Our responses are given in a point-by-point manner below, while all the changes to the manuscript are highlighted in yellow. Regarding the molecular mechanisms provoking the formation of ROS, and the mediators of oxidative stress-induced cellular damage in stroke, we included additional information in the introduction.

“In general, the presentation of the studies that evaluated the strength of oxidative stress biomarkes in ischemic stroke is sufficient and very well-done. The references cited are up-to-date, but also include seminal studies that are not so recent. The authors have already published original research articles related to oxidative stress and the role of antioxidants in disease prevention in high-impact journals. Hence, their expertise is considered sufficient.”

Response: The manuscript content and reference list were revised and where the general information didn’t need any seminal studies endorsement, the references were removed.

“However, the authors state that this is a synthetic review. I would consider it as a narrative review.”

Response: We changed ”synthetic” to ”narrative”. Thank you for your observation.

“Line 25: severity-reflecting instead of severity-dependent?”

Response: We have corrected to “severity-dependent”.

“Line 31: before “thus” end the previous sentence. I think it is too long already.”

Response: The sentences were corrected accordingly.

“Lines 35-37: this sentence seems to me like a repetition of the previous one.”

Response: We revised the meaning of the two sentences and one of them was removed.

“Page 1, line 45: renders them instead of makes them?”

Response: “Makes” was replaced by “renders”.

“Page 2, line 47: pathogenic mechanisms instead of pathological?”

Response: “Pathological” was replaced by “pathogenic”.

“Line 54: research instead of re-search”

Response: The typo was eliminated.

“Line 61: risk factors for the ischemic stroke”

Response: The abnormal inversion of terms was corrected.

“Line 67: being instead of are”

Response: “Are” was replaced by “being”.

“Lines 114-117: I do not understand what you are trying to express. Do you mean that energy depletion precedes oxidative stress?”

Response: In this paragraph we aimed to describe the energy metabolism impairments that precede the cascade of oxidative stress injury that includes oxidation/antioxidation imbalance in the body, and consequently excessive production of ROS and hydroxyl radicals. However, we revised the paragraph and concluded that it is hard to follow. Thus, we made some changes to it, as it follows:

“A link between energy metabolism depletion, oxidative stress, and ischemic stroke has been proposed. The interruption of blood flow to the brain contributes to defective energy metabolism in the pathophysiological hallmark of ischemic stroke. This impaired energy metabolism initiates the cascade of oxidative stress injury characterized by an oxidant/antioxidant imbalance in the body, and consequently excessive production of ROS and hydroxyl radicals, resulting in brain damage which follows a stroke.”   

“General comment: please be more precise about the molecular mechanisms leading to ROS formation in stroke.”

Response: As stated before, additional information regarding this aspect was included in the introduction.

“Lines 128-132: how did you conclude that the existence of cell-free mitochondrial DNA is a marker of increased ROS? Circulating mitochondrial DNA is increased due to tissue damage in general.”

Response: We are truly sorry for this omission. We found that a recent case-control study showed that decreased mtDNA in the peripheral leukocytes could significantly indicate the occurrence of an ischemic stroke event. Also, 8-hydroxy-2'-deoxyguanosine (as one of the predominant forms of ROS molecular lesions) levels were significantly increased in patients with ischemic stroke, as compared with controls (acc. to Omari et al., 2021). However, since correlation does not necessarily mean causation, we revised the paragraph and eliminated the confusing information. 

“You seem to use the words diagnostic and prognostic instead of diagnosis and prognosis”

Response: The manuscript was revised for appropriate use of these terms and “diagnostic” and “prognostic” were replaced “diagnosis” and “prognosis” throughout the manuscript.

“Figure 1: some parts of the figure are not readable. Why do you say “damage cytoskeleton” and then in the brackets you write all the types of macromolecules?”

Response: We revised Figure 1. We replace “damage cytoskeleton” with damage to the enumerated molecules.

“Line 179: the definition of ferroptosis is by too simplistic.”

Response: We revised the definition (according to ÄŒepelak I, et al. 10.2478/aiht-2020-71-3366):

“Ferroptosis is an iron dependent form of non-apoptotic programmed cell death, caused by the harmful accumulation of lipid-based reactive oxygen species due to the failure of the complex lipid peroxide repair systems, such as glutathione-GPX4.”

“Table 1 is very well structured and provides really meaningful information”

Response: Thank you for your kind words of appreciation.

“Why did the authors selectively discuss about myeloperoxidase?”

Response: We chose to selectively discuss myeloperoxidase contribution considering the future perspectives it brings as a potent therapeutic target and the activity it shares between oxidative stress and neuroinflammation in the pathological process of ischemic stroke. Also, extensive experimental and clinical data has shown a correlation between the overexpression of myeloperoxidase and the risk for stroke. It has also been demonstrated that myeloperoxidase and its active products are involved in the onset and development of hemorrhagic and ischemic stroke, and therefore can be used for clinical evaluation and prognosis of stroke (according to Wang et al. Neural Regeneration Research, 2022, 17(8), 1711 and Chen et al. Frontiers in physiology, 2020, 11, 433). These aspects were also explained in the myeloperoxidase section to endorse its role as a potential therapeutic target.

“Please, use shorter sentences. It has been really hard to follow.”

Response: We thoroughly read the manuscript and revised the excessively long sentences.

“I believe the authors should emphasize that oxidative stress biomarkers can be used mainly on monitoring the severity and prognosis. Their role in diagnosis is unlikely; increased ROS are implicated in many diseases. According to this review, it does not seem to exist a marker to discriminate among these diseases.”

Response: Thank you for your valuable suggestion. A paragraph was added in the end of the section 5.1:

“Despite the important role of oxidative stress in the pathogenesis of ischemic stroke, diagnosing ischemic stroke using oxidative stress markers remains a significant challenge since oxidative stress occur in many diseases associated with energy metabolism impairment, neuroinflammation, tissue damage, and cell loss. However, the relevance of oxidative stress markers could still be endorsed in monitoring the ischemic stroke severity and prognosis.”

Thank you for all your help in improving the quality of our manuscript.

Sincerely,

Reviewer 2 Report

The manuscript “The Role of Potential Oxidative Biomarkers in the Prognosis of Acute Ischemic Stroke and the Exploration of Antioxidants as Possible Preventive and Treatment Options” is devoted to a relevant topic, but in my opinion its quality is still insufficient for publication.

The review does not use the classification of stroke by etiological subtypes, although it plays a key role in understanding its pathogenesis. The role of sulfur-containing compounds as biomarkers is not reflected in any way. It would be appropriate to supplement Sections 2–4 with images of biomarker structures and schemes of their generation pathways. Finally, if the review is devoted to markers potentially used in diagnostics, attention should be paid to methods for determining these markers.

l.46 “However, many studies have previously validated some animal models of strokes that enabled the description of pathological mechanisms, risk factors, and possible management options.” I don't understand the meaning of the first part of the sentence.

l.52 “ Despite that it is accepted that strokes are caused by clots or stenosis (ischemia) or blood vessels hypertension (aneurysm), the more recent re-search in this area revealed their multifactorial aetiology, thus admitting the different prognosis and outcomes of affected subjects [5,6].” The term multifactorial aetiology itself is controversial, since there can be only one etiology of the disease.

l.53 “Ischemic strokes often occur because of a thromboembolic event within the cerebral blood flow that leads to neurologic function loss due to vessel dysfunction (atherosclerotic, fibromuscular dysplasia, inflammatory condition, arterial dissection) [1].” Do the authors mean to say that “atherosclerotic, fibromuscular dysplasia, inflammatory condition, arterial dissection” are thromboembolic events?

l. 56. “ In some cases, while the thromboembolic event could be occurring in a different part of the body, the produced clots are transported by the blood flow to the brain where ischemia occurs [7,8].” What prevented the authors from immediately classifying the etiological subtypes of stroke and not confusing readers with some unknown “a different part of the body”?

l.59 “ It was suggested that the ischemic strokes risk factors differ by etiologic subtypes …”. The classification of risk factors is not related to the etiology.

l.119 “As an immediate consequence, reperfusion occurs [25].” What is the result of reperfusion?

l.120 “At the same time, the great amount of energy that is consumed in this process leads to proportional accumulation of H+ and H2O2.” How, through what systems and where does this accumulation take place?

l.124 “Moreover, the presence of ROS excess could influence cerebral blood flow, by stimulating vasoconstriction, platelet aggregation, and endothelial cell permeability [27].” However, at the beginning of this paragraph, the authors wrote that ROS production causes reperfusion.

l. 126 “Also, a recent case-control study showed that the presence of mitochondrial DNA in the peripheral leukocytes could significantly indicate the occurrence of an ischemic stroke event [28]” That is, in the absence of stroke, the mitochondria of the peripheral leukocytes somehow function without DNA? The study [28] made a completely different conclusion: “Our results demonstrate that low mtDNA content in peripheral blood leukocytes is associated with ischemic stroke.”

l. 131 “oxidative stress extent could very much compel prognosis and outcomes following an acute ischemic stroke related to cognitive impairments and mortality risk” In this case, it was a good idea to quantify this and indicate how the oxidative stress extent is measured.

l.155 “As many studies have previously reported relevant oxidative stress biomarkers solely in blood samples collected following the thrombolytic therapy due to ethical reasons, [21,29,31]. There is limited understanding regarding the role of these specific biomarkers in the events occurring just before or during a stroke event.” Obviously this is one sentence.

l.249 “However, since HNE concentration can be determined based on immunological techniques using anti-HNE antibodies, it is preferred over MDA detection [61].” Provide a rationale for this statement

Table 1: 4-HNE (μmol) -> 4-HNE (μmol/L)

l.502 “Based on the degradation (chain breaks) of hydrophilic and lipophilic compounds in the presence of 0.5M perchloric acid, perchloric acid oxygen radical absorbance capacity (ORACPCA) assay could provide information on the antioxidant capacity [42].” Is it true that it is in the presence of perchloric acid that the degradation of hydrophilic and lipophilic compounds occurs? Is perchloric acid a catalyst or a participant in these processes? What chemical reactions involving perchloric acid contribute to ORACPCA?

l.505 “ORACTOT represents the antioxidant capacity of the remaining small molecular weight compounds, its values being generally lower than ORACPCA”. It is obvious that these terms are confused here. “Total ORAC (ORACTOT) represents the antioxidant capacity of plasma; precipitation of plasma proteins by 0.5 M perchloric acid (ORACPCA) in a 1:1 ratio with the sample reflects the antioxidant capacity of the remaining small-molecular-weight compounds” [Lorenzano S, et al. Neurology. 2019;93(13):e1288-e1298].

l.511 “ total oxygen radical absorbance capacity (ORACTOT)” . The interpretation of this term has already been given above.

Author Response

Reviewer 2

“The manuscript “The Role of Potential Oxidative Biomarkers in the Prognosis of Acute Ischemic Stroke and the Exploration of Antioxidants as Possible Preventive and Treatment Options” is devoted to a relevant topic, but in my opinion its quality is still insufficient for publication.”

Response: Thank you very much for all your notes, time, effort, and support in improving our manuscript. We have carefully read the comments and have revised the manuscript accordingly. You can find our point-to-point responses to your suggestions and all the changes in the manuscript are highlighted in yellow. 

“The review does not use the classification of stroke by etiological subtypes, although it plays a key role in understanding its pathogenesis. The role of sulfur-containing compounds as biomarkers is not reflected in any way. It would be appropriate to supplement Sections 2–4 with images of biomarker structures and schemes of their generation pathways.”

Response: The classification of stroke by etiological subtypes and details regarding the subtypes specific pathogenesis were added in Introduction (lines 50-57).

“According to their pathogenic mechanisms, strokes are commonly divided into two subtypes: ischemic and hemorrhagic. Despite that it is accepted that strokes are caused by clots or stenosis (ischemia) or blood vessels hypertension (aneurysm), the more recent re-search in this area revealed that the vast majority of stroke have multifactorial etiology, thus admitting the different prognosis and outcomes of affected subjects [5,6]. Ischemic strokes often occur because of thrombus (in situ occlusion of an artery), an embolism (thrombus of cardiac sources), hypoperfusion (systemic or local low blood flow), or a combination of these [7].”

Also, we have added a sulfur-containing compounds (methionine sulfoxide) as biomarkers of oxidative stress to the section 4.

“Methionine sulfoxide can also be considered an in vivo biomarker of oxidative stress. It is a major derivative of methionine oxidation with ROS by a two-electron dependent mechanism [120] Balasubramanian et al. [121] reported an association between the methionine sulfoxide and the increased risk of incident stroke. Another study conducted by Li et al. [122], shown that the increase in methionine sulfoxide reductases enzymes, which catalyze the conversion of methionine sulfoxide to methionine, following betaine administration had a neuroprotective impact in ischemia/reperfusion injury. As methionine oxidation was found to contribute to cerebral ischemia/reperfusion injury through the potentialisation of NF-κB–dependent adhesion molecule activation [123], it could be relevant to consider methionine sulfoxide as a potential marker of oxidative stress in acute ischemic stroke.”

“Finally, if the review is devoted to markers potentially used in diagnostics, attention should be paid to methods for determining these markers.”

Response: The role and significance of oxidative stress markers in ischaemic stroke and the possibility of their use in the diagnosis as well as the appropriate methods to evaluate them make the subject of its own study that is currently in hypothesis stage in our group. However, currently oxidative stress is yet a very common impairment in many diseases. Since no specific oxidative stress marker was previously described for ischemic stroke, the description of assay methods would be rather common to other conditions.

l.46 “However, many studies have previously validated some animal models of strokes that enabled the description of pathological mechanisms, risk factors, and possible management options.” I don't understand the meaning of the first part of the sentence.

Response: Thank you for your comment, the sentence was revised:

“The life-threatening character of strokes renders them major medical emergencies and the treatment must be administered without delay. This may be one of the factors that make it difficult to collect data from human subjects and observe the pathogenic mechanisms that occur immediately before and after the stroke. The use of animal models has made it possible to mimic the stroke processes, allowing for the description of pathological mechanisms, risk factors, and potential management options. As a result, researchers have valuable information on ischemic stroke.”

l.52 “Despite that it is accepted that strokes are caused by clots or stenosis (ischemia) or blood vessels hypertension (aneurysm), the more recent research in this area revealed their multifactorial aetiology, thus admitting the different prognosis and outcomes of affected subjects [5,6].” The term multifactorial aetiology itself is controversial, since there can be only one etiology of the disease.

Response: We are hereby citing the references that we read when we wrote this sentence:
“A very small proportion of strokes are attributable to monogenic conditions, the vast majority being multifactorial, with multiple genetic and environmental risk factors of small effect size.” This sentence is part of the study Chauhan, G., & Debette, S. (2016). Genetic risk factors for ischemic and hemorrhagic stroke. Current cardiology reports, 18(12), 124.).

l.53 “Ischemic strokes often occur because of a thromboembolic event within the cerebral blood flow that leads to neurologic function loss due to vessel dysfunction (atherosclerotic, fibromuscular dysplasia, inflammatory condition, arterial dissection) [1].” Do the authors mean to say that “atherosclerotic, fibromuscular dysplasia, inflammatory condition, arterial dissection” are thromboembolic events?

Response: In a thrombotic event, the blood flow to the brain is obstructed. This obstruction may be caused by a dysfunction within the vessel itself. This dysfunction is often secondary to atherosclerotic disease, arterial dissection, fibromuscular dysplasia, or inflammatory condition (Hui et al 2018). In order to prevent any confusion to the reader, the sentence was revised:

“Ischemic strokes often occur because of thrombus (in situ occlusion of an artery), an embo-lism (thrombus of cardiac sources), hypoperfusion (systemic or local low blood flow), or a combination of these [7]. In some cases, these events occur within the cerebral blood flow leading to neurological functions loss. Blood vessels dysfunctions are usually the result of atherosclerosis, fibromuscular dysplasia, inflammatory conditions, or arterial dissection [1].”

  1. 56. “In some cases, while the thromboembolic event could be occurring in a different part of the body, the produced clots are transported by the blood flow to the brain where ischemia occurs [7,8].” What prevented the authors from immediately classifying the etiological subtypes of stroke and not confusing readers with some unknown “a different part of the body”?

Response: To avoid any uncertainties in expressing our ideas, information regarding the classification of strokes was included in this paragraph:

“According to their pathogenic mechanisms, strokes are commonly divided into two subtypes: ischemic and hemorrhagic. Despite that it is accepted that strokes are caused by clots or stenosis (ischemia) or blood vessels hypertension (aneurysm), the more recent re-search in this area revealed that the vast majority of stroke have multifactorial etiology, thus admitting the different prognosis and outcomes of affected subjects [5,6]. Ischemic strokes often occur because of thrombus (in situ occlusion of an artery), an embolism (thrombus of cardiac sources), hypoperfusion (systemic or local low blood flow), or a combination of these [7].”

l.59 “It was suggested that the ischemic strokes risk factors differ by etiologic subtypes …”. The classification of risk factors is not related to the etiology.

Response: The sentence was corrected into: “The risk factors of ischemic strokes are usually divided into non-modifiable risks (age, sex, race, or ethnicity, genetic predisposition, and low weight at birth), and modifiable risk factors (diabetes mellitus, alcohol abuse, obesity, hypertension, smoking, metabolic syndrome, dyslipidaemia syndrome, and others)”.

l.119 “As an immediate consequence, reperfusion occurs [25].” What is the result of reperfusion?
l.120 “At the same time, the great amount of energy that is consumed in this process leads to proportional accumulation of H+ and H2O2.” How, through what systems and where does this accumulation take place?

l.124 “Moreover, the presence of ROS excess could influence cerebral blood flow, by stimulating vasoconstriction, platelet aggregation, and endothelial cell permeability [27].” However, at the beginning of this paragraph, the authors wrote that ROS production causes reperfusion.
Response: The paragraph that comprise lines 119-124 was revised alongside a bigger section of the Introductory chapter so that it would further include several additional information regarding the mentioned aspects and the one suggested in the previous observations.

  1. 126 “Also, a recent case-control study showed that the presence of mitochondrial DNA in the peripheral leukocytes could significantly indicate the occurrence of an ischemic stroke event [28]” That is, in the absence of stroke, the mitochondria of the peripheral leukocytes somehow function without DNA? The study [28] made a completely different conclusion: “Our results demonstrate that low mtDNA content in peripheral blood leukocytes is associated with ischemic stroke.”

Response: We are truly sorry for this omission. This part was also revised (please see manuscript). 

l.131 “oxidative stress extent could very much compel prognosis and outcomes following an acute ischemic stroke related to cognitive impairments and mortality risk” In this case, it was a good idea to quantify this and indicate how the oxidative stress extent is measured.

Response: New sentence was added at line 180: “For an instance, Wang et al [50] found that the correlation between oxidized LDL plasma levels and cognitive performance following stroke (as evaluated by Mini-Mental State Examination test) fit into a linear regression model. The same group also reported that stroke patients with increased oxidized LDL plasma levels had higher risk to die or to have poor functional outcomes at 1 year after a stroke event, when this event was related with large-artery atherosclerosis [51,52].”

l.155 “As many studies have previously reported relevant oxidative stress biomarkers solely in blood samples collected following the thrombolytic therapy due to ethical reasons, [21,29,31]. There is limited understanding regarding the role of these specific biomarkers in the events occurring just before or during a stroke event.” Obviously this is one sentence.

Response: Thank you, we revised the sentence.

l.249 “However, since HNE concentration can be determined based on immunological techniques using anti-HNE antibodies, it is preferred over MDA detection [61].” Provide a rationale for this statement.

Response: New reference was added: Schmidt et al (1996). Pediatric Research, 40(1), 15-20. In this study, the authors stated that: “HNE is a more specific parameter for estimation of lipid peroxidation processes in comparison with MDA”, while [61] stated that “Currently, the measurement of MDA is based on the test of thiobarbituric acid (TBA), a substance which can react with various compounds in the body fluid. Moreover, TBA itself can lead to the generation of extra MDA; therefore, an overestimation of the MDA level seems inevitable. Thus, future studies need to innovate more accurate test methods, facilitating its role as a biomarker of strokerelated thrombosis.” and “In contrast to MDA, HNE can be detected through immunological techniques with anti-HNE antibodies or high-performance liquid chromatography (HPLC) directly.” (Li, Z.; Bi, R.; Sun, S.; Chen, S.; Chen, J.; Hu, B.; Jin, H. The role of oxidative stress in acute ischemic stroke-related thrombosis. Oxid Med Cell Longev. 2022, 2022, 8418820.) This is the reason why immunological assays are better than TBA (for MDA) and also why HNE is preferred, despite MDA is more abundant than HNE.

“Table 1: 4-HNE (μmol) -> 4-HNE (μmol/L)”

Response: Revised. Thank you.

l.502 “Based on the degradation (chain breaks) of hydrophilic and lipophilic compounds in the presence of 0.5M perchloric acid, perchloric acid oxygen radical absorbance capacity (ORACPCA) assay could provide information on the antioxidant capacity [42].” Is it true that it is in the presence of perchloric acid that the degradation of hydrophilic and lipophilic compounds occurs? Is perchloric acid a catalyst or a participant in these processes? What chemical reactions involving perchloric acid contribute to ORACPCA?

Response: The information was verified and corrected according to the literature.

“The ORACPCA assay provides a direct measure of antioxidant capacity of the serum nonprotein fraction treated with perchloric acid (PCA) [152].”

l.505 “ORACTOT represents the antioxidant capacity of the remaining small molecular weight compounds, its values being generally lower than ORACPCA”. It is obvious that these terms are confused here. “Total ORAC (ORACTOT) represents the antioxidant capacity of plasma; precipitation of plasma proteins by 0.5 M perchloric acid (ORACPCA) in a 1:1 ratio with the sample reflects the antioxidant capacity of the remaining small-molecular-weight compounds” [Lorenzano S, et al. Neurology. 2019;93(13):e1288-e1298].

Response: Thank you for your attention in reading our manuscript. As a result to your suggestion, we corrected the mentioned aspect:

“ORACTOT, a biomarker of plasma; precipitation of plasma proteins antioxidant capacity, by 0.5 M perchloric acid (ORACPCA) in a 1:1 ratio with the sample reflects the antioxidant capacity of the remaining small-molecular-weight compounds (ORACTOT values are generally lower than ORACPCA) [46]”

l.511 “total oxygen radical absorbance capacity (ORACTOT)”. The interpretation of this term has already been given above.

Response: We revised the use of the abbreviation.

Thank you for all your help in improving the quality of our manuscript.

Sincerely,

Round 2

Reviewer 2 Report

The authors of the manuscript “The Role of Potential Oxidative Biomarkers in the Prognosis of Acute Ischemic Stroke and the Exploration of Antioxidants as Possible Preventive and Treatment Options” made a number of corrections, but some of my comments were ignored and substantive answers were not received.

1. The review does not use the classification of stroke by etiological subtypes, although it plays a key role in understanding its pathogenesis. Despite the statements of the authors, there is no classification of stroke subtypes in the review; references here [5-7] are not relevant. An example is the review [Stroke 1993;24:35-41] or later works.

2. The role of sulfur-containing compounds as biomarkers is reflected insufficiently. Although the authors reflected the role of methionine sulfoxide, aminothiols (glutathione, homocysteine, S-adenosylhomocysteine) are not mentioned in the review. Also Methionine sulfoxide was not listed in Table 1.

3. It would be appropriate to supplement Sections 2–4 with images of biomarker structures and schemes of their generation pathways. This comment was ignored.

4. If the review is devoted to markers potentially used in diagnostics, attention should be paid to methods for determining these markers. I do not fully agree with the authors’ answer here (“The role and significance of oxidative stress markers … make the subject of its own study”), since the authors have already outlined the goal of the review, the achievement of which requires taking into account the capabilities of modern diagnostic tools (“In this narrative review we aimed to identify how the alterations of oxidative stress biomarkers could suggest a severity-reflecting diagnosis of ischemic stroke…”). 

For example, the statement “However, since HNE concentration can be determined based on immunological techniques using antiHNE antibodies, it is preferred over MDA detection [72].” (l. 280) can only be justified at the level of methodology, according to the authors' extended response to my earlier comment. Therefore, I would like to see the appropriate editing of the manuscript so that readers do not have the same question.

Author Response

Letter to Reviewer 2

Dear Reviewer,

Thank you very much for all your notes, time, efforts, and support in improving our paper; we have carefully read the comments and have revised/ completed the manuscript accordingly. Our responses are given in a point-by-point manner below (in blue), as well, all the changes to the manuscript are highlighted in yellow. To improve the quality of the manuscript, the text was modified, completed, corrected, and restructured. 

""The authors of the manuscript “The Role of Potential Oxidative Biomarkers in the Prognosis of Acute Ischemic Stroke and the Exploration of Antioxidants as Possible Preventive and Treatment Options” made a number of corrections, but some of my comments were ignored and substantive answers were not received.

1. The review does not use the classification of stroke by etiological subtypes, although it plays a key role in understanding its pathogenesis. Despite the statements of the authors, there is no classification of stroke subtypes in the review; references here [5-7] are not relevant. An example is the review [Stroke 1993;24:35-41] or later works.

Thank you for observation, we have corrected it as requested

Based on the multicenter Trial of Org 10172 in Acute Stroke Treatment (TOAST) Clas-sification System, the ischemic stroke subtypes are categorized as follows: (1) cardioembo-lism (linked to cardiac dysrhythmias, valvular heart disease, and left ventricle thrombi), (2) large-artery atherosclerotic, (3) lacunar (microatheromatosis), (4) other specific etiology (dissections, vasculitis, specific genetic disorders, and others), and (5) strokes of unknown etiology [6, 7].

  1. The role of sulfur-containing compounds as biomarkers is reflected insufficiently. Although the authors reflected the role of methionine sulfoxide, aminothiols (glutathione, homocysteine, S-adenosylhomocysteine) are not mentioned in the review. Also Methionine sulfoxide was not listed in Table 1.

Thank you for your comment. We have completed section 4 with the requested biomarkers, as follows :

 Homocysteine

Homocysteine (2-amino-4-mercaptobutyric acid) is a non-proteinogenic sul-fur-containing amino acid derived from demethylation of methionine via S-adenosylmethionine (SAM) and S-adenosylhomocysteine (SAH) [126]. Homocysteine metabolism represents an intersection of two pathways: 1) remethylation pathway regen-erating methionine via methylfolate homocysteine methyltransferase (methionine syn-thase), its coenzyme, methylcobalamin, and 5-methyltetrahydrofolate (MTR) or beta-ine-homocysteine methyltransferase (BHMT) as the methyl donor ; 2) transsulfuration pathway generating cystathionine via cystathionine β-synthase (CBS), the cystathionine thus formed is then transformed into cysteine via cystathionine γ-lyase (CTH); both en-zymes require the cofactor pyridoxal phosphate (vitamin B6) [126, 127]. Both prospective and retrospective investigations have revealed a link between plasma homocysteine levels and the risk of ischemic stroke [128-131]. Epidemiological data suggest that homocysteine is a significant predictor of ischemic stroke. Moderately elevated homocysteine values (even in the population reference range) are linked to vascular pathology by a variety of mechanisms, including atherosclerotic and thrombotic events [130]. Patients with is-chemic stroke, both in the acute and convalescent phases, have been found to have hy-perhomocystéinémie [132, 133]. A nested case-control study from the Netherlands showed that a 1 µmol/L increase in total homocysteine concentrations was associated with a 6% to 7% increase in stroke risk. The study also showed an increased risk of stroke above the highest quintile (18.6 µmol/L) compared with the lowest quintile (12.0 µmol/L), with an odds ratio of 2.43 (1.11 to 5.35) [129]. Similarly, Perry et al. [128] found that among mid-dle-aged British men, stroke patients had considerably higher serum total homocysteine concentrations (across quartiles) (geometric mean 11.9 [11.3-12.6] µmol/L) than in controls (geometric mean 13.7 [12.7-14.8] mol/L). In comparison to the first quarter, there was a gradual quartile rise in the relative stroke risk in the second, third, and fourth quarters of the total homocysteine distribution (odds ratios 1-3, 1-9, 28; trend p=0-005) [128]. Con-sistent with prior research in older people, Niazi et al. [134] showed that 50.7% of young ischemic stroke patients exhibited a moderate to high frequency of Homocysteine. They also found that homocysteine levels were markely higher in men (mainly in the 36-45 age group) than in women [134]. Perry et al. [128] bots et al. [129] and Niazi et al. [134] char-acterized hyperhomocysteinemia (HHcy) as a values > 12μmol/L. These three investiga-tions reported a significant correlation between HHcy and stroke incidence, implying that HHcy can be used as a predictive risk factor for stroke progression [128, 129, 134]. Fur-thermore, Zhao et al. assessed the ability of S-adénosylhomocystéine hydrolase (AHCY), an enzyme responsible for catalysis of S-adenosylhomocysteine into adenosine and ho-mocysteine, in predicting the outcomes of IS [135]. The study showed that the percentage of DNA methylation of AHCY was significantly (p < 0.0001) higher in patients with is-chemic stroke (0.13% (0.09%, 0.27%) than in controls (0.06% (0.00%, 0.17%) [135].

4.8. Glutathione

Glutathione (γ-L-Glutamyl-L-cysteinylglycine) is a low molecular weight sul-fur-containing pseudo-tripeptide formed by the condensation of glutamic acid, cysteine and glycine. It is the major non-protein thiol in many tissues.  It is involved in various vital processes, including redox homeostatic buffering [136]. It participates in the protec-tion of protein thiol groups against oxidative damage and the neutralization of reactive oxygen species. Glutathione exists in two forms: the reduced form (GSH) and the oxidized form (GSSG). Over 90% of total glutathione (GSH) is in the reduced form (GSH). In the brain, glutathione is mainly in a reduced form in high concentration (~1-3 mM) [136-138]. In two experimental rat models of ischaemia caused by hypoperfusion (middle cerebral artery occlusion and bilateral occlusion of the common carotid arteries), GSH homeostasis as an oxidative stress marker has been disturbed in global and focal ischaemia. A drop in reduced and a significant increase in oxidized GSH was shown in the brain on rat models of cerebral ischemia after MCAO and BCAO. This effect may be attributed to the fact that the dynamic GSH homeostasis shifts toward the oxidized form (GSSG) as a result of oxi-dation [139]. In human study, Ozkul et al. [140] reported an increase in serum GSH levels in 70 subjects within 48 hours after stroke compared with matched controls [140].  Simi-lar to this, Zimmermann et al. [141] noted an increase in GSH and GPX levels during the first 6 hours and 1 day after the acute stroke, respectively, when compared to controls [141]. According to these results, the increased GSH levels could be part of first-line de-fense mechanisms against oxidative stress and could provide adaptive mechanisms to oxidative stress during AIS [140, 141].

  1. It would be appropriate to supplement Sections 2–4 with images of biomarker structures and schemes of their generation pathways. This comment was ignored.

Please accept our apologies for that and thank you for your valuable suggestion, the figures have now been added :

Figure 2. MDA generation cycle: (1) the high reactivity of polyunsaturated fatty acids to oxidizing agents is promoted by the methylene groups positioned between the cis double bonds along the chain. The removal of the hydrogen atoms leads to unsaturated lipid carbon-centered radicals (2) vulnerable to oxidizing and conversion into highly unstable peroxyl radicals (3). Depending on the position of the peroxyl radical in the carbon chain, two pathways are possible: terminal position at one of the two ends of the double bond system will result in reduction to conjugated diene hydroperoxides (4) while internal position will result in multiple products by mono- and/or bi-cyclizations, coupling to O2 or reductions. The fragmentation of a bicyclic peroxide compound (5) structurally analogous to the prostaglandin endoperoxide generates the malondialdehyde (6).

Figure 3. The myeloperoxidase (MPO) catalytic cycle. Ferric myeloperoxidase (MPO-Fe3+ in its resting state) reacts reversibly with hydrogen peroxide by oxidation of the heme group and formation of the ferryl-p cation radical intermediate compound I (1). This intermediate can oxidize halides and the pseudo-halide thiocyanate to the corresponding hypohalous acids, while being readily reduced to MPO-Fe3+ (2). Compound I also oxidizes multiple organic substrates (RH) to free radicals (R.) in the classical peroxidase cycle which involves two sequential one electron reduction steps, generating the compound II (3) and III (4). Peroxidation substrates include tyrosine estradiol, serotonin, norepinephrine, ascorbate, urate etc. Superoxide can also be the substrate for compound I and compound II in these stages. Ferric myeloperoxidase can be reduced to its ferrous form MPO-Fe2+, which binds to O2, forming the oxymyeloperoxidase or compound III (MPO-Fe2+O2). Superoxide also provides the substrate for the fast conversion of the ferric enzyme to form compound III (5). Ferrous myeloperoxidase and compound III can display catalytic action by reductively activating quinones (adapted from Podrez et al, [153]).

  1. If the review is devoted to markers potentially used in diagnostics, attention should be paid to methods for determining these markers. I do not fully agree with the authors’ answer here (“The role and significance of oxidative stress markers … make the subject of its own study”), since the authors have already outlined the goal of the review, the achievement of which requires taking into account the capabilities of modern diagnostic tools (“In this narrative review we aimed to identify how the alterations of oxidative stress biomarkers could suggest a severity-reflecting diagnosis of ischemic stroke…”).

    For example, the statement “However, since HNE concentration can be determined based on immunological techniques using antiHNE antibodies, it is preferred over MDA detection [72].” (l. 280) can only be justified at the level of methodology, according to the authors' extended response to my earlier comment. Therefore, I would like to see the appropriate editing of the manuscript so that readers do not have the same question."""

Thank you very much for your valuable comment; a table of  detection strategy of oxidative stress biomarkers has been added in section 6, as follows :

Table 2. Detection strategy of oxidative stress biomarkers

Oxidative Stress Biomarkers

Detection Strategy Examples

References

Malondialdehyde

UV-Vis

Fluorescence

Electrochemistry

GC-MS

SERS

TBARS Test

[175, 176]

F2-isoprostanes

GC-MS

GC-NICI-MS

ELISA

HPLC-MS/MS

SPE-HPLC-MS/MS

[177-180]

4-Hydroxy-2-Nonenal

2-AP

HPLC

Sandwich ELISA

Western Blot

LC-MSMS

FT-ICR MS

MALDI-TOF-MS

32P-PostlabelingGC-MS

DNPH Derivatization

[181-183]

Oxidised LDL

TBARS assay

ELISA

[184, 185]

8-oxo-7,8-dihydro-2-deoxyguanosine

HPLC-ED

[186]

3-Nitrotyrosine

(HPLC)-(UV-VIS) absorption Electrochemical (ECD)

Diode array (DAD)

LC-MS

LC-MS/MS

GC-MS

GC-MS/MS

Sandwich ELISA

[187,188]

Gluthatione

DTNB/GR enzyme recycling method

HPLC

[189]

Protein carbonyls

Western blot

In-gel fluorophoric tagging

Levine spectrophotometric method

ELISA

HPLC

[190, 191]

Homocysteine

HPLC with fluorometric detection

HPLC-ED

Immunonephelometric method

LC-MS-MS

Fluorescence polarization

Immunoassay

EIA

[192]

Methionine sulfoxide

Peptide mapping with MS detection

rpHPLC

HIC

Weak cation-exchange chromatography

[193]

Myeloperoxidase

BLI

CRET 

ADHP

MPO-Gd MR imaging

[194, 195]

GC-MS : gas chromatography-mass spectrometry ; NICI-MS : gas chromatography-negative-ion chemical ionization mass spectrometry ; HPLC : high performance liquid chromatography ; SPE : polymeric weak anion-exchange solid-phase extraction ; UHPLC–MS/MS : isotope-dilution ultrahigh performance liquid chromatography electrospray ionization–tandem mass spectrometry ; 2-AP : fluorescent probe 2-aminopyridine ; DNPH : 2,4-dinitrophenylhydrazine ; HPLC-ED : high performance liquid chromatography with electrochemical detection ; EIA : enzyme-linked immunoassay ; BLI : bioluminescence imaging ; CRET : chemiluminescence resonance energy transfer ; MPO-Gd : bis-5-hydroxytryptamide-diethylenetriaminepentaacetate-gadolinium ; ADHP : 10-acetyl-3,7-dihydroxyphenoxazine ; TBARS : thiobarbituric Acid Reactive Substances.

Thank you for all your help in improving the quality of our manuscript.

Sincerely,
